# Modeling the Glacial Lake Outburst Flood Process Chain in the Nepal Himalaya: Reassessing Imja Tsho's Hazard

Jonathan M. Lala[1], David R. Rounce[2], and Daene C. McKinney[1]

[1]Center for Water and the Environment, University of Texas at Austin, Austin, TX, USA
[2]Geophysical Institute, University of Alaska Fairbanks, Fairbanks, AK, USA

*Correspondence to*: Jonathan M. Lala (jonalala@hotmail.com)

**Keywords:** numerical model, avalanche, GLOF, inundation, process chain modeling, hazard assessment

**Abstract.** The Himalayas of South Asia are home to many glaciers that are retreating due to climate change and causing the formation of large glacial lakes in their absence. These lakes are held in place by naturally deposited moraine dams that are
potentially unstable. Specifically, an impulse wave generated by an avalanche or landslide entering the lake can destabilize the moraine dam, thereby causing a catastrophic failure of the moraine and a glacial lake outburst flood (GLOF). Imja-Lhotse Shar glacier is amongst the glaciers experiencing the highest rate of mass loss in the Mount Everest region, in part due to the expansion of Imja Tsho. A GLOF from this lake may have the potential to cause catastrophic damage to downstream villages, threatening both property and human life, which prompted the Nepali government to construct outlet
works to lower the lake level. Therefore, it is essential to understand the processes that could trigger a flood and quantify the potential downstream impacts. The avalanche-induced GLOF process chain was modeled using the output of one component of the chain as input to the next. First, the volume and momentum of various avalanches entering the lake were calculated using RAMMS. Next, the avalanche-induced waves were simulated using BASEMENT and validated with empirical equations to ensure the proper transfer of momentum from the avalanche to the lake. With BASEMENT, the
ensuing moraine erosion and downstream flooding was modeled, which was used to generate hazard maps downstream. Moraine erosion was calculated for two geomorphologic models: one site-specific using field data and another worst-case based on past literature that is applicable to lakes in the greater region. Neither case resulted in flooding outside the river channel at downstream villages. The worst-case model resulted in some moraine erosion and increased channelization of the lake outlet, which yielded greater discharge downstream but no catastrophic collapse. The site-specific model generated
similar results, but with very little erosion and a smaller downstream discharge. These results indicated that Imja Tsho is unlikely to produce a catastrophic GLOF due to an avalanche in the near future, although some hazard exists within the downstream river channel, necessitating continued monitoring of the lake. Furthermore, these models were designed for ease and flexibility such that local or national agency staff with reasonable training can apply them to model the GLOF process chain for other lakes in the region.

## 1. Introduction

The Hindu Kush - Himalaya Region contains more glacial ice and perennial snow than any other region on Earth outside the polar regions, and supplies water via its rivers to over a fifth of the earth's population (Qiu, 2008; Matthew, 2013). While these glaciers are undeniably significant in sustaining the populations of South and East Asia, they also provide some of the best gauges for understanding regional and global climate change, since temperatures in high altitudes are increasing faster than in lower elevations (Wang et al., 2017; Kraaijenbrink et al., 2017). Mass loss on both debris-covered and clean-ice glaciers has been observed throughout the Himalayas, and glacial lake formation has been increasing since the 1960s (Bolch et al., 2008; Nie et al., 2017). For glaciers where the surface slope is small and surface velocity is slow ($< 10$ m a$^{-1}$), meltwater and precipitation tend to pool in small ponds, which act as a heat sink for solar radiation and accelerate glacial melt (Quincey et al., 2007; Mertes et al., 2016). Eventually, these small ponds can coalesce and become the large glacial lakes found throughout the Hindu Kush - Himalaya Region (Benn et al., 2012).

While glacier mass loss due to climate change is a long-term water resource problem (Kraaijenbrink et al., 2017), the formation of large glacial lakes poses a more immediate threat to local populations. Two thirds of the glacial lakes in Nepal are held in place by natural moraine dams, which are potentially prone to failure (ICIMOD, 2011). Events such as an avalanche or landslide entering a glacial lake can cause tsunami-like waves that could overtop and/or erode these moraines and trigger a glacial lake outburst flood (GLOF). The ensuing GLOF could have a devastating impact on both property and human lives downstream. Furthermore, climate change is exacerbating glacial retreat and mass loss, making a major avalanche event more likely (Schneider et al., 2011). In the Mount Everest region alone (Figure 1), glacier-wide mass loss averages around 0.52 m w.e. a$^{-1}$, with the surface of Imja-Lhotse Shar Glacier losing an average of 1.56 m w.e. a$^{-1}$—the largest mass loss in the region, due to the accelerated melt caused by Imja Tsho ("Tsho" meaning "lake" in Tibetan) (King et al., 2017; Thakuri et al., 2016).

Imja Tsho, which has formed at the terminus of Imja-Lhotse Shar Glacier, has been considered one of Nepal's highest-priority lakes for mitigation studies due to its size and proximity to populated areas. In November 2016, it was the subject of a lake lowering project by the Nepalese Army (BBC World Service, 2016). The lake itself is retained by a terminal moraine to the west, bounded by lateral moraines to the north and south, and connected to the glacier to the east along the calving front (Figure 1). While calving from the glacier could cause some small wave generation, the most common cause of GLOFs is an avalanche-generated tsunami wave (Emmer and Cochachin, 2013; Falátková, 2016). At Imja Tsho, hanging ice from the surrounding mountains is too far away to affect the lake at the present time (Rounce et al., 2016; Figure 1). However, if the lake continues expanding eastwards towards the surrounding mountains at its current rate, avalanches could potentially enter the lake in the future (Rounce et al., 2016). Therefore, it is important to model these potential avalanches and determine if they could initiate a chain reaction of overtopping waves, erosion and subsequent discharge at the terminal moraine. Modeling this chain is particularly important for Imja Tsho, as an outburst flood might result in the loss of lives and property at communities like Dingboche, which is only 8 km downstream.

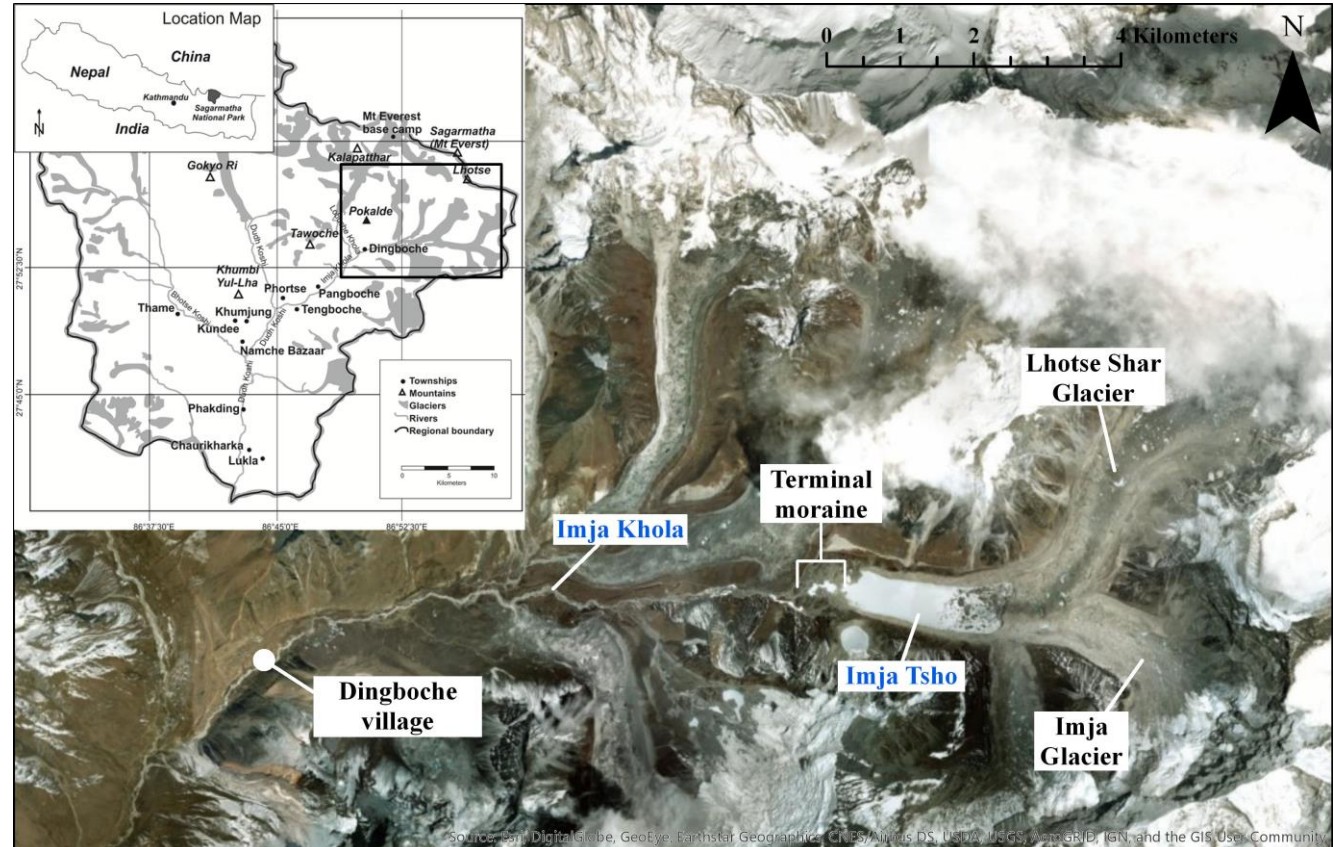

**Figure 1: Overview of study area showing the Mount Everest region (inset, Nicholson et al. (2016)), Imja Tsho, primary glaciers, avalanche-prone hanging ice (dark blue, Rounce et al. 2016), and the Imja Khola channel down to Dingboche village. Source: DigitalGlobe, Inc. (imagery); inset reprinted from Nicholson et al. (2016) under Creative Commons Attribution International License.**

The most important morphological feature that contains the lake is the terminal moraine, composed of boulders, gravel, and sand. The moraine is relatively wide, extending approximately 600 meters westward from the lake. The outlet of the lake (Figure 2) consists of a series of ponds surrounded by hummocky terrain that could potentially reduce the risk of a GLOF by absorbing energy and storing water from an overtopping wave (Hambrey et al., 2008). The size of the moraine would likely prevent a wave from completely overtopping it; however, a wave could still scour the outlet channel and lead to rapid discharge, threatening communities downstream.

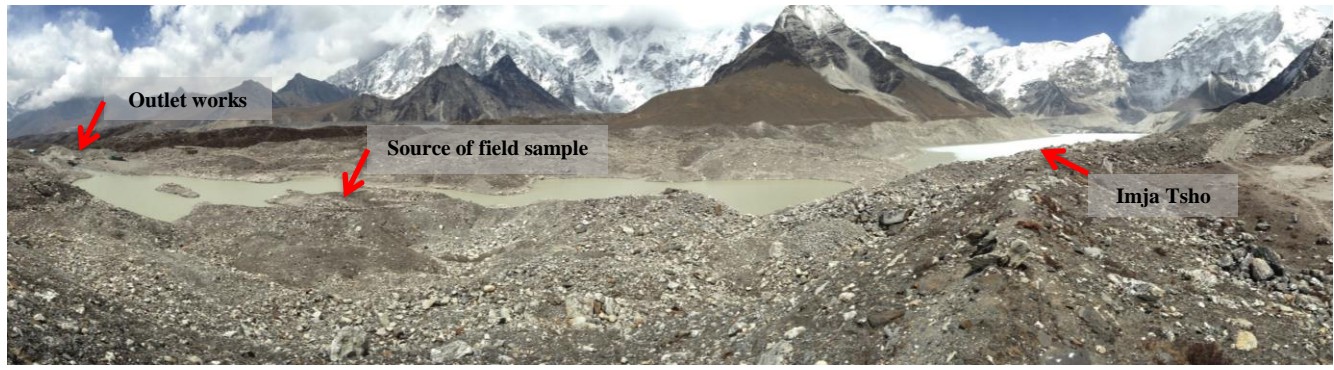

**Figure 2: Imja Tsho's terminal moraine and outlet pond complex showing the lake water (right side of image) flowing westward through a series of ponds to the outlet of the terminal moraine (left side of image) (photo acquired 27 April 2017). The location of a field sample used in this study is also shown.**

In spite of evidence that avalanches are the most common trigger of GLOFs in the Himalayas (Falátková, 2016), previous hazard assessments of Imja Tsho have largely relied on assumptions concerning the breach of the moraine as opposed to modeling it through a realistic process chain. Somos-Valenzuela et al. (2015) computed inundation at the downstream village of Dingboche for various lake surface lowering scenarios, but assumed dam breaching was caused by piping resulting from slow melting of the ice core within the damming moraine and specified the dimensions and timing of the breach.

Bajracharya et al. (2007) similarly assumed dam breaching due to a melting of the moraine's ice core as well as a decrease in the width of the moraine. Shrestha and Nakagawa (2016) modeled inundation scenarios for an overtopping event, but did not model wave processes in the lake or the overtopping wave causing the moraine erosion. Furthermore, hazard assessments of Imja Tsho that are not based on numerical or experimental modeling (i.e. those based on remote sensing and in-situ surveys) have had mixed results, with some indicating high hazard (Kattelmann, 2003; ICIMOD, 2011; Somos-

Valenzuela et al., 2015), low hazard (Hambrey et al., 2008; Fujita et al., 2009; Watanabe et al., 2009), or a moderate hazard at the present and high hazard in the future (Rounce et al., 2017).

Studies at other lakes in the Mount Everest region have similarly relied on unverified assumptions. Cenderelli and Wohl (2001) and Dwivedi (2007) did not include debris flow or erosion in their models, even though such factors are major contributors to downstream inundation (Osti and Egashira, 2009). Shrestha et al. (2013) included debris flow in a GLOF

model of Tsho Rolpa, and assumed moraine failure from both seepage and overtopping; however, overtopping was due to a steady rise in the lake level and not due to an impulse wave. Recent studies have yielded more complex models regarding multiphase debris flows such as the open-source r.avaflow, which can simulate an avalanche-induced GLOF process chain in a single model, but this model is still in development and has yet to be calibrated by observed real-world data (Mergili et al., 2017). A replicable process chain model for avalanche-induced GLOFs is therefore greatly needed to assess GLOF hazard

throughout the Himalayas.

This study seeks to employ a comprehensive set of models to evaluate the present and future hazard associated with avalanche-generated impulse waves at Imja Tsho. This model chain represents an easily replicable method that can be applied to other lakes. Specifically, this study addresses all components of the GLOF process chain, including:

1. Avalanche generation and propagation,
2. Wave generation, propagation, and runup, and
3. Moraine erosion and subsequent downstream flooding.

Understanding these components will assist in the wider goal of helping local communities adapt to the risks associated with glacier recession, increasing the capacity for climate change resilience.

## 2. Methods

Glacial lake hazards are determined from a variety of climatic and geographic factors, of which increasing climate variability is paramount, since it reduces the stability of glaciers, snowpack, and bedrock and hence increases the frequency of avalanches (Fischer et al., 2012). In areas like the Nepal Himalaya, where the climate is warming, the topography is steep, and there is an abundance of seismic activity, an avalanche is the most common GLOF trigger (Emmer and Cochachin, 2013; Falátková, 2016). Since avalanche-induced GLOFs are a chain of individual events, there are generally two options for characterizing them in the absence of true integrative modeling: modeling each component and using their outputs as inputs for the next component in the chain, or approximating components so that the chain can be simulated in a single model run (Worni et al., 2014). The methodology used in this study presents a hybrid approach using two models: modeling the avalanche in a single model, and then using its output as the input for an environmental flow modeling software that takes into account the subsequent wave, moraine erosion, and downstream debris flow and inundation.

### 2.1 Avalanche Modeling

Impulse waves generated by mass movement into lakes are common in alpine regions, where avalanches can be large and impact velocities can be high (Heller et al., 2009), hence, avalanches are the most common GLOF triggering mechanism in the Himalayas (Emmer and Cochachin, 2013; Falátková, 2016). For a realistic avalanche-triggered GLOF scenario to be computed, the source and trajectory of an avalanche must first be determined.

Ice and snow cover near Imja Tsho was previously identified by Rounce et al. (2016) with Landsat imagery using a ratio of NIR (near-infrared) and SWIR (short wave infrared) bands with a threshold of 2.2 (Huggel et al., 2004a). Any ice-covered area with a slope between 45° and 60° was considered avalanche-prone (Figure 1); slopes above this limit are generally too steep to allow for mass accumulation (Alean et al., 1985; Osti et al., 2011). Finally, the areal extent of the initial block of mass to be released was determined using a variable kernel filter, grouping avalanche-prone pixels together if 90% of the surrounding pixels are also avalanche-prone (Rounce et al., 2016). Ice thickness ranges were determined based on observations in Russia (Huggel et al., 2005), standard values in Switzerland (Huggel et al. 2004b), and estimates in the

Chinese Himalaya (Wang et al., 2012). Assumed values fell between 10 and 50 m, such that when combined with areal extents, the total avalanche volume could reach from $2.7 \times 10^4$ to $6.7 \times 10^6$ m$^3$ (Rounce et al., 2016).

Avalanches were modeled using Rapid Mass Movements Simulation (RAMMS) Debris Flow Module (Bartelt et al., 2013). RAMMS uses the Voellmy-Salm finite volume method to solve the depth-averaged equations governing mass flow in two

dimensions, with second-order accuracy (Christen et al., 2010). RAMMS can also model entrained material in a mass flow, which makes it useful for GLOF simulations (Worni et al., 2014). The basic required inputs for RAMMS include a digital elevation model (DEM), the initial avalanche release area and its depth, and parameters for debris density and friction. The Voellmy-fluid friction model used in RAMMS requires two friction parameters: $\mu$ and $\xi$, the velocity-independent dry-Coulomb and velocity-dependent turbulent friction terms, respectively (Bartelt et al., 2013). For the case study presented

here, values of $\mu = 0.12$, $\xi = 1000$ m s$^{-2}$, and $\rho = 1000$ kg m$^{-3}$ were used, which agree with values used in previous GLOF-producing avalanche models (Schneider et al., 2014; Somos-Valenzuela et al., 2016). A sensitivity analysis of these values indicates that they are conservative, since they produce the fastest, farthest-traveling, and densest avalanches within accepted standard values (see Bartelt et al., 2013).

## 2.2 Field Surveys and Future Lake Extents

A bathymetric survey of the Imja Tsho was conducted on 16-17 June 2016 using an inflatable kayak and a Garmin echoMAP 54dv to measure 4399 points of lake depth. The lake's shoreline was manually delineated using a clear-sky WorldView-02 image (DigitalGlobe, Inc.) from 14 May 2016. The shoreline was converted into point measurements and combined with the depth measurements to interpolate the depth over the entire lake using the Topo-to-Raster tool in ArcGIS. The lake depth raster was then burned into a regional DEM with a resolution of ~ 4 m (King et al., 2017). This DEM covered the area

between the lake and Dingboche and was used as an input for the models in this study.

Currently, there is no realistic avalanche scenario that can enter the lake; however, the lake is expanding eastwards such that it will be within an avalanche trajectory around 2035 (Rounce et al., 2016; see also Table 5 in Results). Therefore, in order to assess the future hazard, it was necessary to predict the future extent and bathymetry of Imja Tsho. Future lake extents were based on Rounce et al. (2016), which used the average decadal rate of expansion based on lake extents from 2000-2015

in conjunction with estimated future overdeepenings identified by GlabTop2 (Linsbauer et al., 2012; Frey et al., 2014). The lake level was assumed to remain constant in future projections, since it has remained relatively constant in the past 15 years (Rounce et al., 2016). The results from the 2016 bathymetric survey were then combined with the overdeepenings identified by GlabTop2 to predict lake bathymetry for future scenarios (Rounce et al., 2016). This future bathymetry was then burned into the DEM.

Although the lake was subjected to a lowering project in the summer of 2016 that reportedly lowered the lake by 3 m (BBC World Service, 2016), this lowering was not accounted for in the GLOF process chain modeling as it is unclear how much the main lake was lowered based on repeat satellite imagery (Figure 3). Specifically, WorldView-2 (0.5 m; DigitalGlobe, Inc.) images of the lake's outlet complex before and after the lowering project show a clear ring of discoloration and

decrease in area around the outlet ponds, but the lack of discoloration near the shore of the main lake suggests that the main lake may not have been lowered to the same extent. Hence, the GLOF process chain was modelled conservatively by not accounting for any lake lowering.

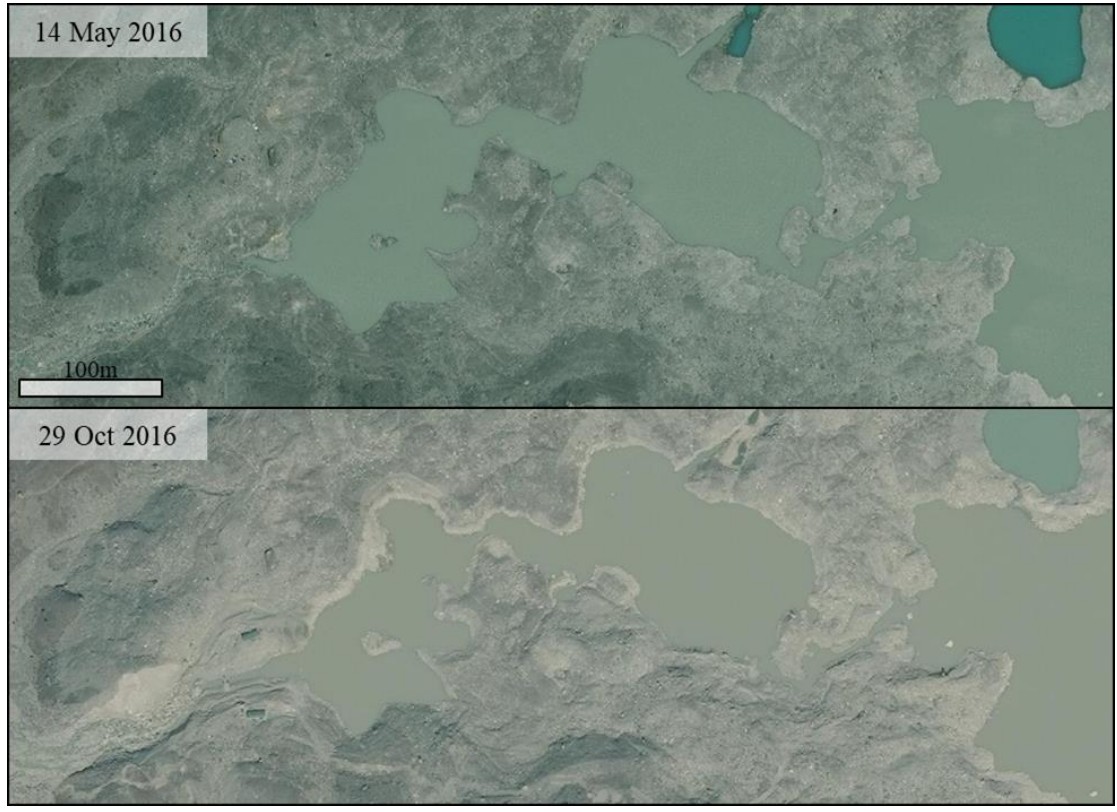

**Figure 3: WorldView-2 (0.5 m; DigitalGlobe, Inc.) imagery of the lake before (14 May 2016) and after (29 October 2016) the lake lowering project, showing a ring of discoloration around the outlet ponds but not the main lake.**

The terminal moraine was also assumed to remain stable in the future. While there is evidence that the moraine has lowered over time (Watanabe et al., 1995), the western shoreline of the lake adjacent to the terminal moraine has remained stable since the late 1980s (Fujita et al., 2009). Furthermore, the moraine's width and gentle slope add to its stability such that

degradation of the ice core or piping will not likely pose a major risk, and a wave is more likely to cut through the outlet rather than completely overtop the moraine (Rounce et al., 2016). Weakening of the terminal moraine due to seismic activity was similarly disregarded based on the moraine's width and the lack of appreciable harm it suffered from the 2015 Gorkha earthquake and the earthquake's aftershocks (Byers et al., 2017).

### 2.3 GLOF Model

Most methods used to characterize waves generated by avalanches into lakes rely on numerical or empirical models, as analytical methods often cannot capture the complexity of subaerial wave generation (Yavari-Ramshe and Ataie-Ashtiani,

2016).  Numerical models generally rely on the 2-D shallow water equations (SWE) or Boussinesq-type equations, whereas empirical models rely on simplified geometries and are best used as validation for complex numerical simulations (Somos-Valenzuela et al., 2016).  While Boussinesq models account for nonlinear effects such as dispersion, their computational cost is higher and their application to real situations often provides no significant benefit over SWE models (Murty and Kowalik,

1993).  Conversely, the simplicity of SWE models allows for inclusion of sediment transport, erosion, and deposition without excessive computational time—an advantage of the Basic Simulation Environment for Computation of Environmental Flow and Natural Hazard Simulation (BASEMENT) model (Vetsch et al., 2017).  This study used BASEMENT for modeling all phenomena in the GLOF process chain downstream of the avalanche.

### 2.3.1 Empirical Wave Model

The Heller-Hager model (Heller et al., 2009) is a combination of analytical and empirical equations that model impulse wave generation, propagation, and runup resulting from mass movement entering a lake.  Although the method relies on simplified assumptions about the geometry of lakes, it has been used to successfully model some real-world events and performs well in characterizing the impulse wave within the lake, which makes it a useful as a calibration measure for more complex hydrodynamic models (Somos-Valenzuela et al., 2016).  Moreover, it is not as susceptible to wave attenuation inherent in 2-

D SWE models such as BASEMENT, making it an ideal calibration measure that is both simple and accurate.  The Heller-Hager model was used only to compare wave heights with BASEMENT results; terminal moraine runup was ignored, owing to Imja Tsho's complex geometry and bathymetry.

The Heller-Hager method was applied using avalanche characteristics (width, thickness, density, and lake entry angle and velocity) from RAMMS to determine the characteristics of the ensuing impulse wave, particularly impulse and wave height

(Heller et al., 2009).  These results were used for calibration, i.e., waves in BASEMENT simulations that were of the same order of magnitude as the Heller-Hager waves were generally accepted as more accurate; however, when they were not, mass entry rates were changed, by altering the inflow hydrograph, to more closely match the Heller-Hager results.  Section 2.3.2 provides more details on the calibration procedure.

### 2.3.2 Hydrodynamic Wave Simulation

The processes following the avalanche event—wave generation and propagation, moraine erosion, and downstream debris flow and inundation—were modeled using BASEMENT.  Its function as both a hydrodynamic model and a sediment transport model makes it well suited to model much of the GLOF process chain (Worni et al., 2014).  BASEMENT solves 2-D SWE in combination with sediment transport equations, primarily the Shields parameters and the Meyer-Peter and Müller (MPM) equations (Shields 1936; Vetsch et al., 2017).  BASEMENT can simulate morphology as either a single grain

(MPM), or as multiple grain sizes with the MPM-Multi equations; the latter includes characterization of hiding and armoring of surfaces not present in the single-grain MPM equations (Vetsch et al., 2017).  This dual modeling capability allows modeling of the moraine erosion and dynamic outlet channel discharge, in addition to the impulse wave in the lake.

BASEMENT requires a DEM in the form of a Triangulated Irregular Network (TIN) rather than a traditional raster DEM, which is often ill-suited for hydrodynamic modeling (e.g. false sinks are less common in TINs since surfaces are sloped, whereas any pixel with a value lower than its surroundings creates a sink in a raster DEM). Therefore, the DEM generated from the regional DEM and bathymetric survey results was further processed in QGIS (QGIS Development Team, 2016) to
create a TIN DEM.

The avalanche hydrograph determined from RAMMS was used as the inflow boundary condition for BASEMENT. For each timestep of the avalanche simulation, RAMMS produces a raster of debris deposition. The inflow rate of debris into the lake was determined by adding the values of all cells, for each raster, that were within the lake boundary. Avalanche material is similar in density to water ($\rho = 1000$ kg m$^{-3}$) (Schneider et al., 2014, Somos-Valenzuela et al., 2016), such that
volume was determined as a 1:1 ratio (i.e., 1 m$^3$ of avalanche material entering the lake corresponds to 1 m$^3$ of water entering at the inflow boundary).

BASEMENT distributes inflow evenly along a user-defined boundary, whereas the avalanche enters the lake at various rates along the shore. Defining the inflow boundary is therefore a critical calibration measure. The center of mass of the avalanche along the lakeshore was chosen as the inflow boundary, and the width of the boundary was set so that wave
heights simulated by BASEMENT agreed with those from the Heller-Hager model. In the case that the determined width produced an unstable result (BASEMENT cannot model inflow velocities exceeding 200 m/s, and it tends to create artificial flow overdrafts if the minimum depth per element is set to less than 0.01 m), or results did not match with the Heller-Hager model, the hydrograph was altered. Generally, this required the inflow volume to be increased and the inflow time to be decreased by the same scale factor, so that momentum could be increased without changing the total volume entering the
lake. If the hydrograph was adjusted, the width was also readjusted to match wave heights with the Heller-Hager model.

### 2.3.3 Moraine Morphology and Erosion

Two erosion models were used in BASEMENT for separate simulations: MPM and MPM-Multi (see above). The MPM-Multi model used soil characteristics from a field sample taken along the edge of the outlet channel (27.9004º N, 86.9089º E; Figure 2) on 27 April 2017. The sample was analyzed for grain size distribution, ATSM D422 (ASTM, 2007), and porosity and density, ASTM D7263 (ASTM, 2009). Because the lakebed likely consists mainly of ice or rock (Somos-Valenzuela et
al., 2014), erosion of the lakebed was disregarded except near the terminal moraine.

The MPM-Multi model simulates hiding and armoring processes that can lead to unrealistically low levels of erosion (Vetsch et al., 2017). The MPM model ignores these processes, and can lead to an overestimation of erosion. A very small grain size, generally the $d_{10}$ value of the soil matrix, was used in the MPM model to create a worst-case scenario for moraine
stability (Somos-Valenzuela et al., 2016). Finally, a correction factor of 2.0 was used in both models to increase the rate of bedload transport. Values between 0.5 (low transport) and 1.7 (high transport) are generally realistic, while a value of 2.0 provides the most conservative estimates (Somos-Valenzuela et al., 2016; Table 1).

Currently, information on soil mechanics for wetted and submerged slopes throughout Nepal is limited, which impedes the application of a generalized worst-case scenario for lake-damming moraines in the Mount Everest region. However, some data are available from localized GLOF modeling studies at sites both within and outside the region, which makes it possible to approximate moraine properties based on field observations. Samples from Tsho Rolpa (27.87° N, 86.47° E), a glacial lake 45 kilometers from Imja Tsho, indicate an internal friction angle of 35° for wetted sediment (Shrestha and Nakagawa, 2014), confirming estimations from earlier field surveys at Imja Tsho (ICIMOD, 2011; Shrestha and Nakagawa, 2016). Outside of Nepal, moraine material at Imja Tsho also bears a strong resemblance to that of Ventisquero Negro, Argentina (see Worni et al., 2012), with maximum slopes around 80°, similar grain size distributions ($d_{10} \approx 1$ mm, $d_{50} = 15\text{-}20$ mm), and a noncohesive, unconsolidated mix of boulders, sand, and gravel, such that failure angles would likely be similar between the two sites. Similarly, studies of glacial lakes in the Peruvian Andes have determined submerged slope failure angles to be between 35° and 40°, similar to that of Tsho Rolpa (Novotný and Klimeš, 2014). Because of the similarities in values between Tsho Rolpa, the Andean studies, and visual inspection of moraine material at Imja Tsho, values from these other studies were used in the BASEMENT simulations. Table 1 summarizes the values taken from these studies as inputs for the soil matrix in BASEMENT.

Table 1: Geomorphic parameters used to define soil matrix of the terminal moraine in BASEMENT simulations

| Parameter | Value | Source |
|---|---|---|
| Sediment transport formula | | |
|     General scenario | MPM | Vetsch et al. (2017) |
|     Imja-specific scenario | MPM-Multi | Vetsch et al. (2017) |
| Diameter $d_{10}$ | 1 mm | Somos-Valenzuela et al. (2016); field sample |
| Density | | |
|     General scenario | 2650 kg m$^{-3}$ | Novotný and Klimeš (2014); Shrestha and Nakagawa (2014) |
|     Imja-specific scenario | 1800 kg m$^{-3}$ | Field sample |
| Porosity | | |
|     General scenario | 40% | General value for spherical grain |
|     Imja-specific scenario | 30% | Field sample |
| Bed Load Factor | 2 | Somos-Valenzuela et al. (2016) |
| Sediment failure angle | | |
|     Dry | 77° | Worni et al. (2012) |
|     Submerged | 36.5° | Novotný and Klimeš (2014) |
|     Deposited | 15° | Worni et al. (2012) |

### 2.3.4 Downstream Impact and Hazard Identification

BASEMENT simulations for Imja Tsho were run for up to 2.6 hours after avalanche entry into the lake, which provided sufficient time to assess the debris flow and inundation at the village of Dingboche, 8 km downstream of the lake outlet. Initial discharge from Imja Tsho was assumed to be negligible, since peak monsoon discharge at Dingboche of 4-6 $m^3$ $s^{-1}$

was less than 4% of the peak discharge from the GLOF flood wave (Rajkarnikar, 2013; see Results, Figure 8). The output from the inundation model was used to measure flood intensity (Table 2), a quantitative measurement based on maximum flow velocity and depth (Somos-Valenzuela et al., 2016). Flood intensity was defined in one of three degrees: (1) high: possible injury to humans or animals inside buildings; possible collapse or heavy damage to buildings; (2) medium: possible injury to humans or animals outside buildings; possible damage to buildings; and (3) low: small possibility of injury to

humans or animals inside or outside buildings; building damage generally superficial.

Hazard classification is defined as the relationship between flood intensity and probability. However, since there is a lack of data regarding avalanche probability, a semi-quantitative likelihood approach was used, based on assumed ice and snow thickness and known surface slopes. Likelihood was defined based on avalanche volume: high for small avalanches ($5 \times 10^4$ $m^3$), medium for medium avalanches ($9 \times 10^5$ $m^3$), and low for large avalanches ($6.6 \times 10^6$ $m^3$). Combined with flood

intensity, this yielded a semi-quantitative hazard identification system (Table 3) based on that of Raetzo et al. (2002).

**Table 2: Flood intensity classification as a function of maximum depth and velocity (Somos-Valenzuela et al., 2016)**

| Flood Intensity | | Maximum velocity (m/s) times maximum depth (m) | | |
|---|---|---|---|---|
| | | > 1.0 | 0.2 - 1.0 | < 0.2 |
| Maximum depth (m) | > 1.0 | High | High | High |
| | 0.2 - 1.0 | High | Medium | Low |
| | < 0.2 | High | Low | Low |

**Table 3: Flood hazard classification based on flood intensity (Table 2) and the semi-quantitative system of Raetzo et al. (2002)**

| Flood Hazard | | Likelihood | | |
|---|---|---|---|---|
| | | High | Medium | Low |
| Intensity | High | High | High | High |
| | Medium | Medium-High | Medium | Medium-Low |
| | Low | Medium | Medium-Low | Low |

## 3. Results

### 3.1 Lake Bathymetry and Future Extents

The bathymetric survey indicated a maximum depth of the main lake of 157.7 ± 1 m (Figure 4), a mean depth of 65.2 ± 1 m, a total volume of 88.0 ± 1.4 × 10⁶ m³ (Table 4), and an area of 1.35 km² ± 0.01 km². In contrast, a previous survey of the

main lake from 2012 had a maximum depth of 116.3 m, a mean depth of 48 m, and a total volume of 61 × 10⁶ m³ (Somos-Valenzuela et al., 2014). The main difference between the two surveys was that the 2012 survey was unable to approach the calving front due to the number of icebergs present at the time and could not accurately measure depths greater than 100 m. This caused the 2012 survey to assume an ice ramp that extended from the middle of the lake to the calving front. The 2016 survey was able to provide more reliable estimates of the lake depth as measurements were made close to the calving front

and the Garmin echoMap 54dv was able to accurately measure the deepest parts of the lake (Figure 4). The 2016 survey therefore shows a much more abrupt change in depth near the calving front, which results in a greater volume, mean depth, and maximum depth over the whole lake. Furthermore, the eastward expansion of the lake has resulted in a steady increase in volume and depth (

Table 4) as the lake expands into overdeepenings of the glacier bed. The bathymetry of the outlet ponds was also measured

but not included in the main lake area. The maximum depth was 15.4 ± 1 m, the mean depth was 5.1 ± 1 m, the area was 0.037 ± 0.002 km², and the total volume was 0.19 ± 0.04 × 10⁶ m³.

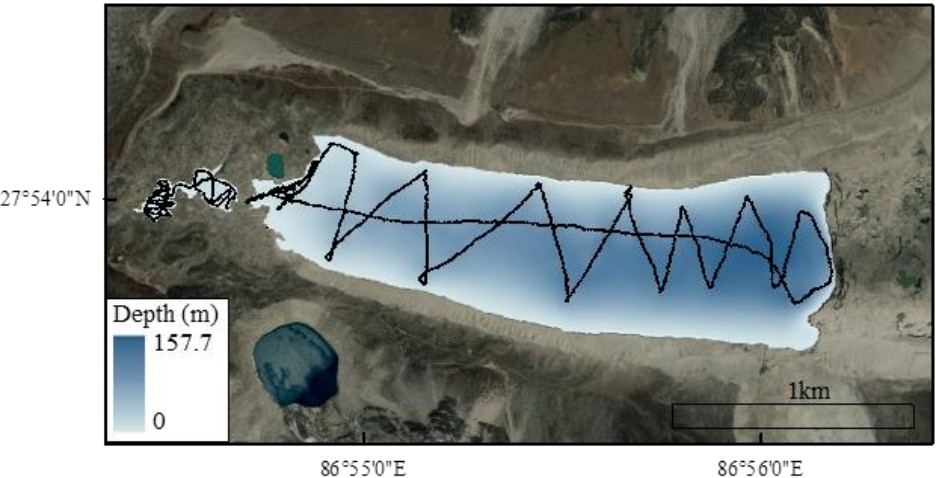

**Figure 4: Bathymetric survey of Imja Tsho from June 2016 showing survey tracks.**

**Table 4: Comparison of 2016 bathymetric survey with previous surveys at Imja Tsho.**

| Survey | No. of points | Total volume ($10^6$ m$^3$) | Avg. depth (m) | Max. depth (m) | Source |
|--------|---------------|------------------------------|----------------|----------------|--------|
| 1992 | 61 | 28.0 | 47.0 | 98.5 | Yamada and Sharma (1993) |
| 2002 | 80 | 35.8 ± 0.7 | 41.6 | 90.5 | Sakai et al. (2003) |
| 2012 | 10020 | 61.7 ± 0.7 | 48.0 ± 2.9 | 116.3 ± 5.2 | Somos-Valenzuela et al. (2014) |
| 2016 | 4399 | 88.0 ± 1.4 | 65.2 ± 1 | 157.7 ± 1 | This study |

Future lake extents were estimated using overdeepenings determined by the GlabTop2 model. The lake expands eastward
for the first 20 years before splitting into two arms extending up the Lhotse Shar Glacier (northeast) and Imja Glacier
(southeast). Figure 5 illustrates these results, which are superimposed by the deposition of two large avalanches (see § 3.2),
one reaching each arm of the future lake.

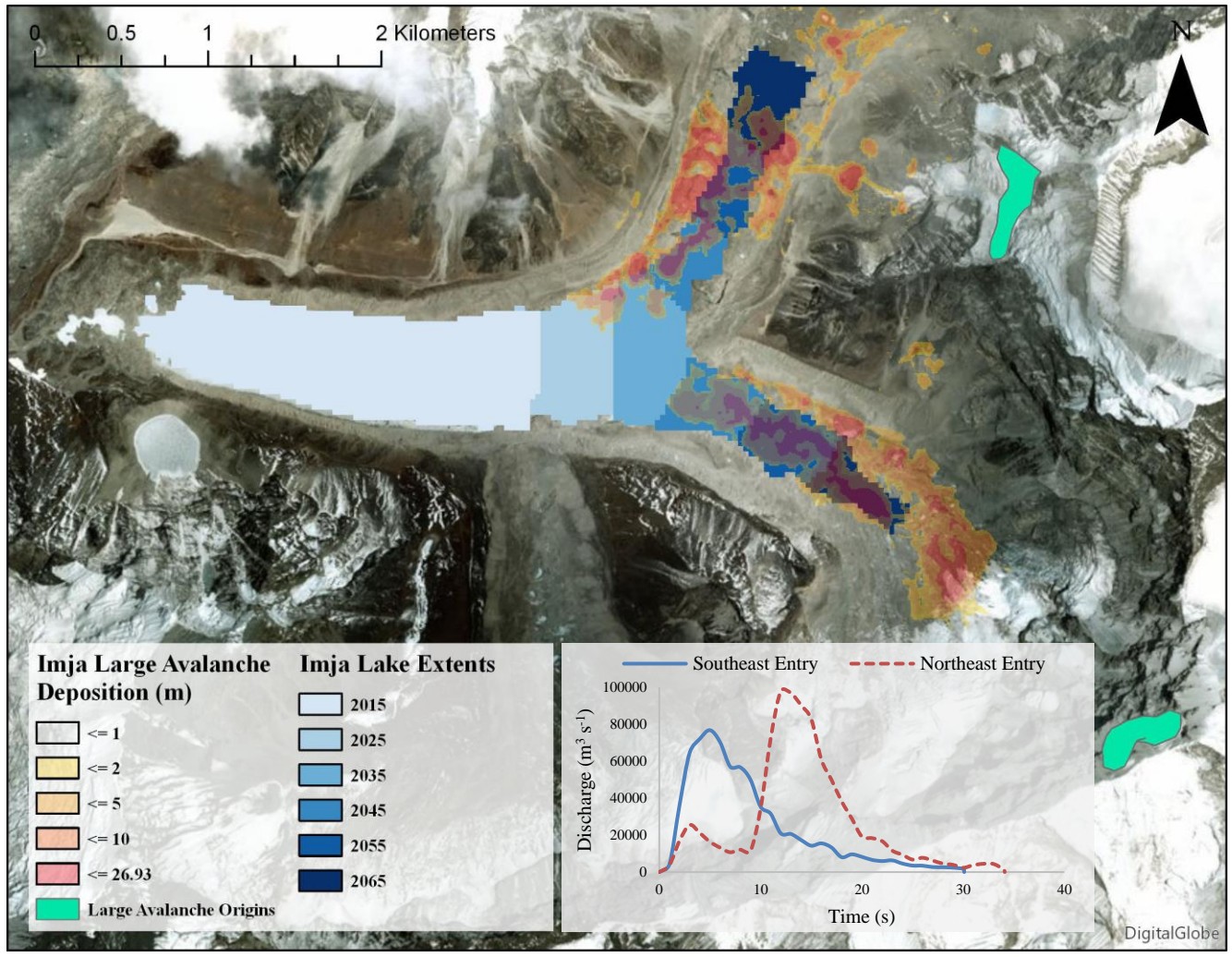

**Figure 5: Deposition from the two large avalanche scenarios superimposed over estimated future lake extents, and time series of avalanche material entry into the lake for the 2045 estimated lake extents (inset). The northeast avalanche enters perpendicularly to the lake expansion trajectory, whereas the southeast avalanche enters with an almost direct trajectory toward the terminal moraine.**

### 3.2 Avalanche Simulations

Avalanche scenarios were computed for two initial starting locations, one to the northeast above the Lhotse-Shar glacier and one to the southeast above the Imja glacier (Figure 5). A small ($5 \times 10^4$ m$^3$), medium ($9 \times 10^5$ m$^3$), and large ($6.6 \times 10^6$ m$^3$) avalanche were considered from both starting locations by varying the areal extent and depth of the initial source released; however, only the large avalanches were able to reach the lake at or before 2045 (Table 5). Specifically, the large avalanche from the northeast reached the lake at the 2025 predicted extent, but mass entry was too small and failed to produce measurable erosion at the terminal moraine unless the predicted 2045 lake was simulated. Large avalanches from both the northeast and southeast reached the lake at the 2045 extent and produced erosion of the moraine. Post-avalanche processes

were analyzed for only the two large avalanche scenarios (northeast and southeast), since these were the only avalanches that reached the lake by 2045 and because they were the only ones to cause measurable erosion at the terminal moraine. The resulting mass entry rates within the lake boundary were used as the inflow hydrographs for subsequent BASEMENT modeling (Figure 5, inset). A comparison of the avalanches from the northeast and southeast showed that the southeast avalanche had a smaller peak discharge into the lake but a larger initial impulse and a steadier decrease in flow. The northeast avalanche had a more variable inflow into the lake, and entered at an angle such that the resulting wave did not propagate directly towards the terminal moraine, which reduced the severity of downstream flooding.

**Table 5: Results for various avalanche scenarios in 2045, showing parameters needed for Heller-Hager model.**

| Avalanche Size | Total volume ($10^6$ m$^3$) | Initial depth (m) | Volume entering lake ($10^5$ m$^3$) | Velocity at lake impact (m s$^{-1}$) | Thickness at lake impact (m) |
|---|---|---|---|---|---|
| Large (Southeast) | 6.6 | 50 | 7.2 | 30 | 24 |
| Large (Northeast) | | | 9.0 | 30 | 26 |
| Medium | 0.9 | 30 | 0 | *n/a* | *n/a* |
| Small | 0.05 | 10 | 0 | *n/a* | *n/a* |

### 3.3 Lake Simulations

In all scenarios, momentum transfer from avalanches into the lake created waves that ran up the terminal moraine, but only large avalanches from 2045 and beyond resulted in sufficient discharge to cause measurable erosion of the moraine at the lake outlet or flooding at Dingboche. The resulting impulse waves were attenuated in the lake, with a reduction of over 80% in the first third of the traverse across the lake due to the rapid increase in lake depth. The wave height stabilized as the lakebed slowly sloped upward toward the lake outlet; finally, runup near the terminal moraine resulted in a slight increase in height (Figure 6). The 2-D SWE in BASEMENT inherently cause the wave to undergo excessive attenuation. Therefore, wave heights were calibrated, by adjusting the inflow hydrographs and boundary widths, so that the amplitudes in both BASEMENT and the Heller-Hager empirical model matched at the far end of the initial wave trajectory (far-field), after the lake depth begins to slope upwards. Although this results in an abnormally high wave near the avalanche entry in BASEMENT, it creates wave heights that closely match that of the Heller-Hager equations at the terminal moraine, which is the focus of this study. Generally, the time from avalanche entry to terminal moraine runup and outlet discharge was approximately 3 minutes; however, the initial trajectory of the wave from the northeast avalanche only lasted approximately 1 minute before it ran up the lateral moraine (Figure 6).

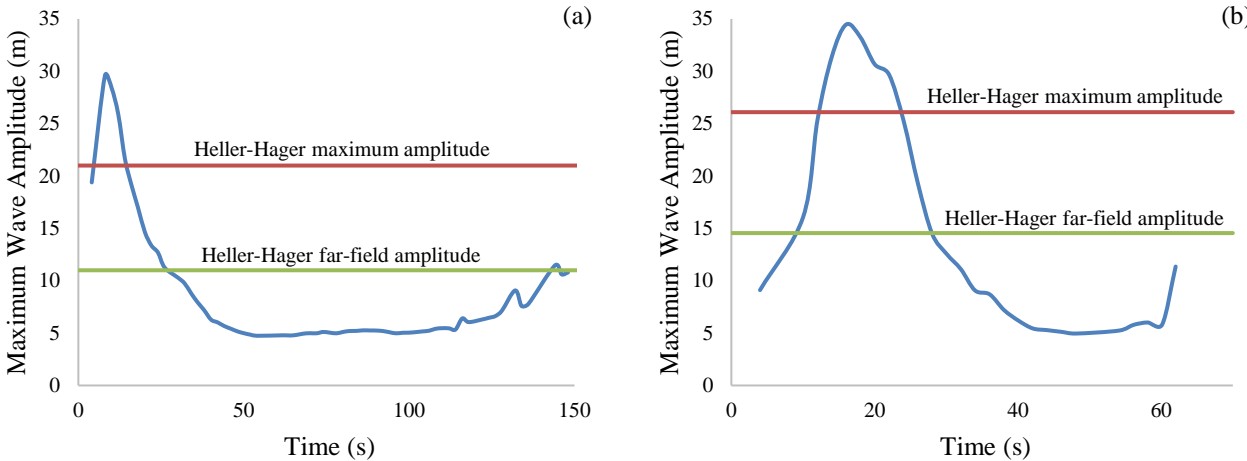

**Figure 6: Maximum amplitude of the leading impulse wave across its initial trajectory, based on a large avalanche entering from southeast (a) and northeast (b) in 2045, showing corresponding wave amplitudes from the Heller-Hager model.**

Wave characteristics for the southeast avalanche scenario (Figure 6a) showed a smaller initial wave height but similar far-field height relative to the northeast scenario (Figure 6b), likely because of the direct line of wave propagation from avalanche entry to the terminal moraine. Conversely, avalanche entry from the northeast arm of the lake resulted in an indirect wave propagation that required some refraction (as the wave approached the south lateral moraine at an angle) and reflection (off of the south lateral moraine) before reaching the terminal moraine (see Figure 5). The resulting loss of energy yielded a smaller runup at the terminal moraine relative to the southeast avalanche scenario and the Heller-Hager results.

## 3.4 Moraine Erosion and Discharge

Erosion and discharge at the terminal moraine were determined for 2000 s (0.5 hr) following avalanche entry into the lake, after which discharge from the lake stabilizes. Three cross sections were analysed: (A) at the lake outlet, where the terminal moraine rises above the lake, (B) at the end of the terminal moraine, and (C) downstream of the terminal moraine within the Imja Khola channel (Figure 7).

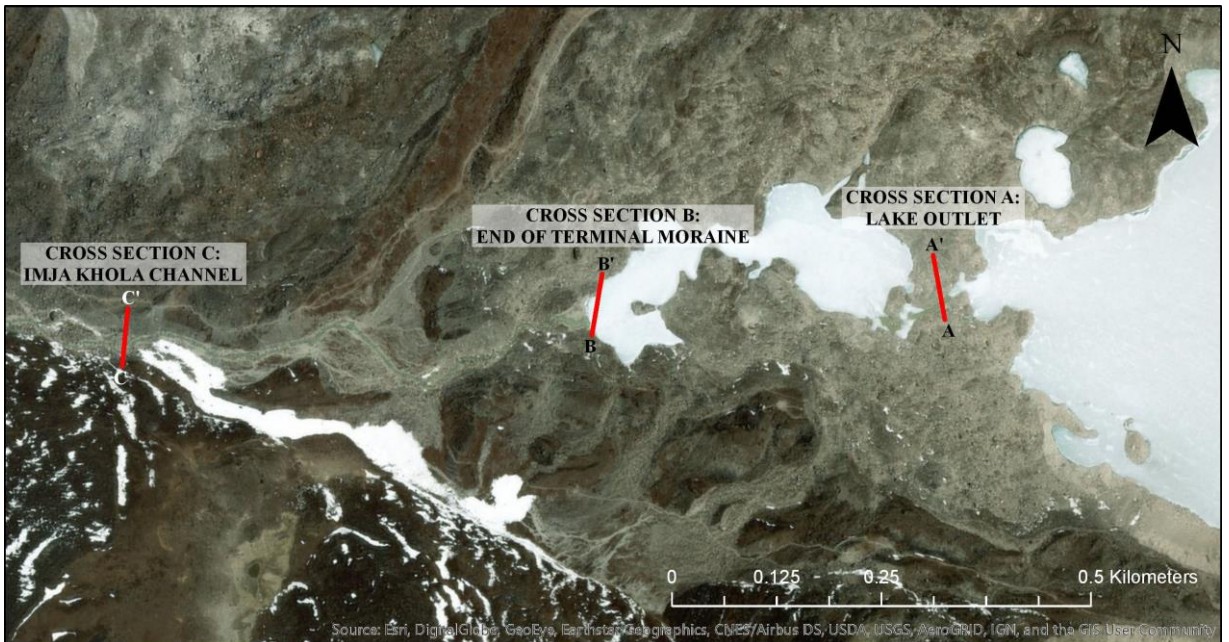

**Figure 7: Location of cross sections for discharge and erosion analysis at the start of the terminal moraine (A), end of the moraine (B), and start of the Imja Khola channel (C).**

Combined sediment and water discharge at the lake outlet (A) for the large southeast avalanche scenario arrived after about 130 s and peaked at 3140 m³ s⁻¹ for the MPM model and 2904 m³ s⁻¹ for the MPM-Multi model (Figure 8). Discharge at the outlet showed considerable oscillation due to the leading and trailing waves caused by the avalanche (Heller et al., 2009). After approximately 900 s, discharge stabilized to around 25 m³ s⁻¹ and erosion ceased. The MPM-Multi model had consistently smaller peak discharges than the MPM model, but a similar oscillatory structure, likely due to the smaller volume of debris within the flow. The flood wave arrived at the end of the moraine (B) after 250 s with a peak 290 m³ s⁻¹ for the MPM model and 134 m³ s⁻¹ for the MPM-Multi model. The discharge here showed less oscillation than at the lake outlet (A). The lower and more stable discharge at (B) suggests that the outlet ponds on the terminal moraine (Figure 2) act as reservoirs that dampen the flood peaks and offer some protection from flooding. After approximately 2000 s, discharge stabilized to around 15 m³ s⁻¹. The MPM-Multi model had consistently smaller peak discharges than the MPM model, again likely because of a lack of sediment transport due to hiding and armouring of the channel, but a similar oscillatory structure and time to stabilization. The flood wave arrived at the Imja Khola channel (C) after 460 s with a peak of 263 m³ s⁻¹ for the MPM model and after 560 s with a peak of 93 m³ s⁻¹ for the MPM-Multi model. The discharge showed less oscillation in the MPM model than at the lake outlet, whereas the MPM-Multi model had dampened all oscillations by this time. After approximately 2000 s, discharge stabilized to around 26 m³ s⁻¹ and 20 m³ s⁻¹ for the MPM and MPM-Multi models, respectively. Debris discharge was also analyzed at Dingboche. For the MPM model, the flood reached the village after 3440 s (almost 1 hr), with a peak discharge of 160 m³ s⁻¹, and steadily decreased afterwards, dipping below 20 m³ s⁻¹ by 9100 s. The MPM-Multi model showed the flood arriving slightly later (4600 s) but discharges were nearly identical and hence

not shown in Figure 8.  In the first 2000 s of the simulation (i.e., before discharge lowers to ~5 m$^3$ s$^{-1}$), the total volume of water leaving the lake was approximately 251,000 m$^3$ for the MPM simulation and 166,000 m$^3$ MPM-Multi simulation—less than 0.3% of the total present lake volume (88 million m$^3$).  This is notably less than the amount of avalanche material entering the lake (approximately 720,000 m$^3$; Table 5).  Moreover, the lake's surface elevation remains slightly above its original elevation at the end of the simulation period (by approximately 0.25 m), suggesting that erosion of the moraine was not sufficient to allow the lake to drain quickly, and may have even allowed the lake to store more water.

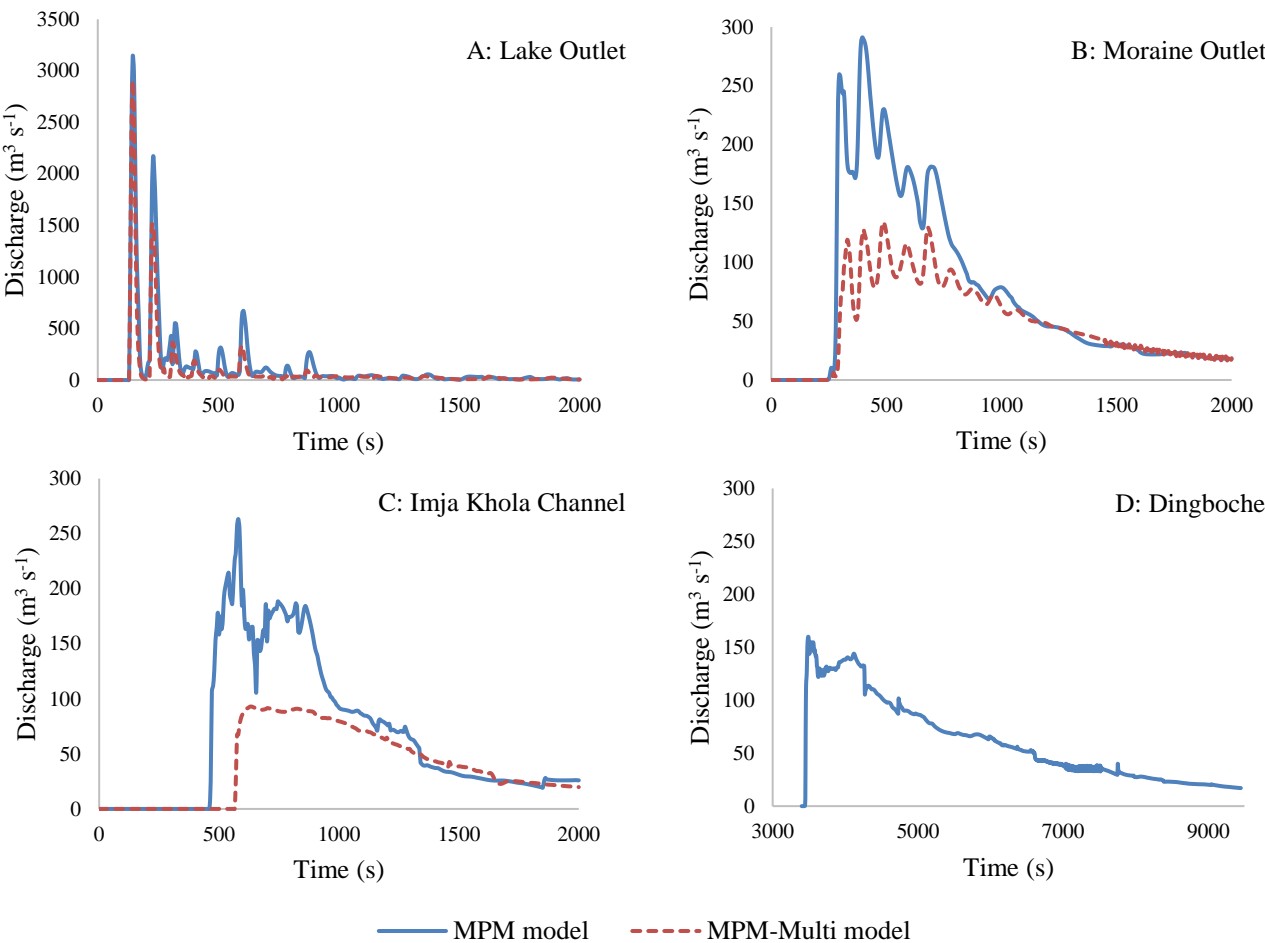

**Figure 8: Combined sediment and water discharge for first 2000 s after initial wave generation from a large southeast avalanche at all three cross sections for both the MPM model (blue line) and the MPM-multi model (red line), plus debris discharge for first 9500 s at Dingboche for the MPM model.  Note the larger discharge scale of (A).**

As expected, the MPM model resulted in more erosion and higher discharge at the terminal moraine.  For all three cross sections, erosion never exceeded 5 m (Figure 9), which is less than the necessary amount needed to reach the ice core of the moraine and accelerate moraine degradation (Hambrey et al., 2008).  The maximum bed erosion at the lake outlet (A) for the

MPM and MPM-Multi models was 4.6 m and 1.7 m, respectively. The maximum bed erosion at the moraine outlet (B) for the MPM model was 0.75 m and for the MPM-Multi model it was negligible. At the Imja Khola channel (C) there was minimal erosion for the MPM model (< 1 m) and negligible erosion for the MPM-Multi model. In both cases, the moraine was not fully overtopped, and erosion was confined to the outlet channel.

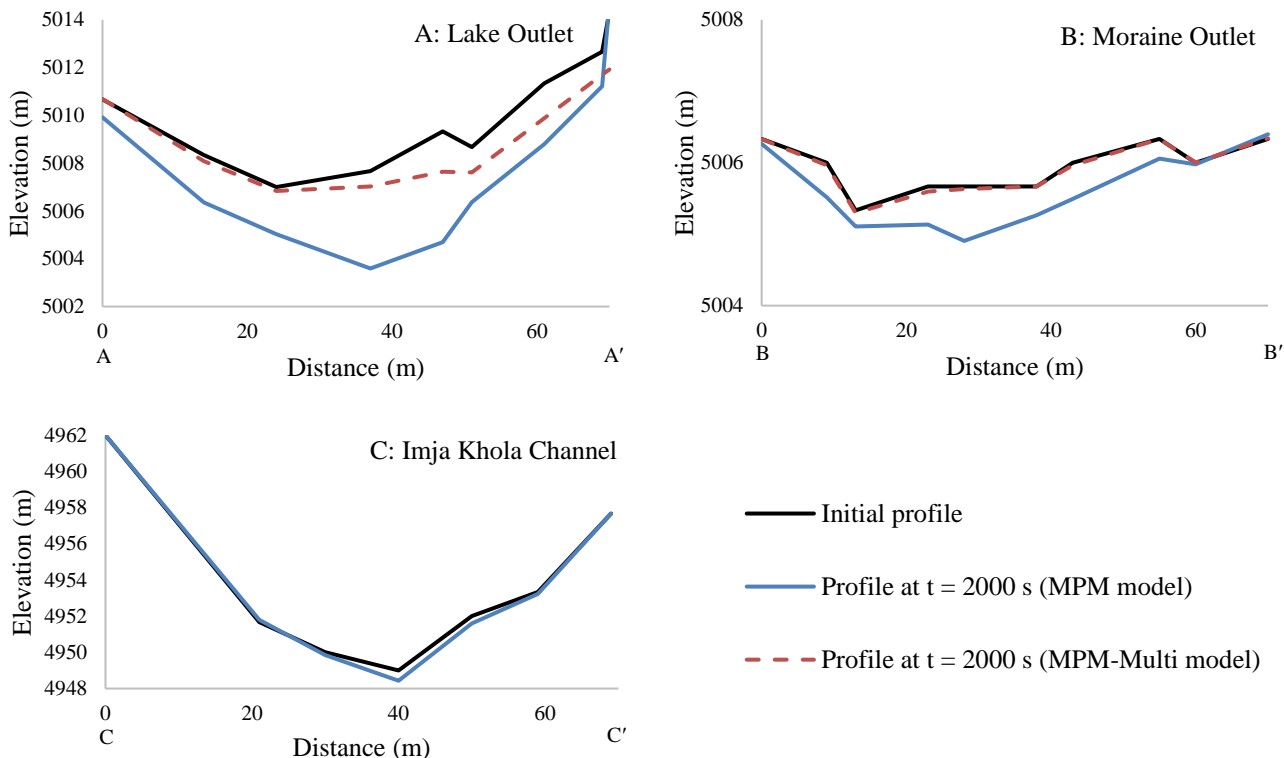

**Figure 9: Surface elevation profiles at the start of wave generation (solid black line) from large southeast avalanche and 2000 s after generation at all three cross sections, for both the MPM model (solid blue line) and the MPM-Multi model (dashed red line), at the lake outlet (A), moraine outlet (B), and Imja Khola channel (C) transects shown in Figure 7  Note the smaller scale of (B). The MPM-Multi model at (C) lacked measurable erosion and is not shown.**

10   **3.5 Downstream Flood Hazard**

In both the MPM and MPM-Multi scenarios, the flood wave reached the village of Dingboche approximately an hour after the avalanche entered the lake. However, floodwater was confined to the river channel in all cases (Figure 10). Scouring of and deposition in the channel near the village was negligible. The maximum flow depth remained less than 3 m at Dingboche and flow velocity did not exceed 6 m s$^{-1}$ (Figure 10). Hazard was therefore negligible in all parts of the village
15   except the river channel (Figure 11).

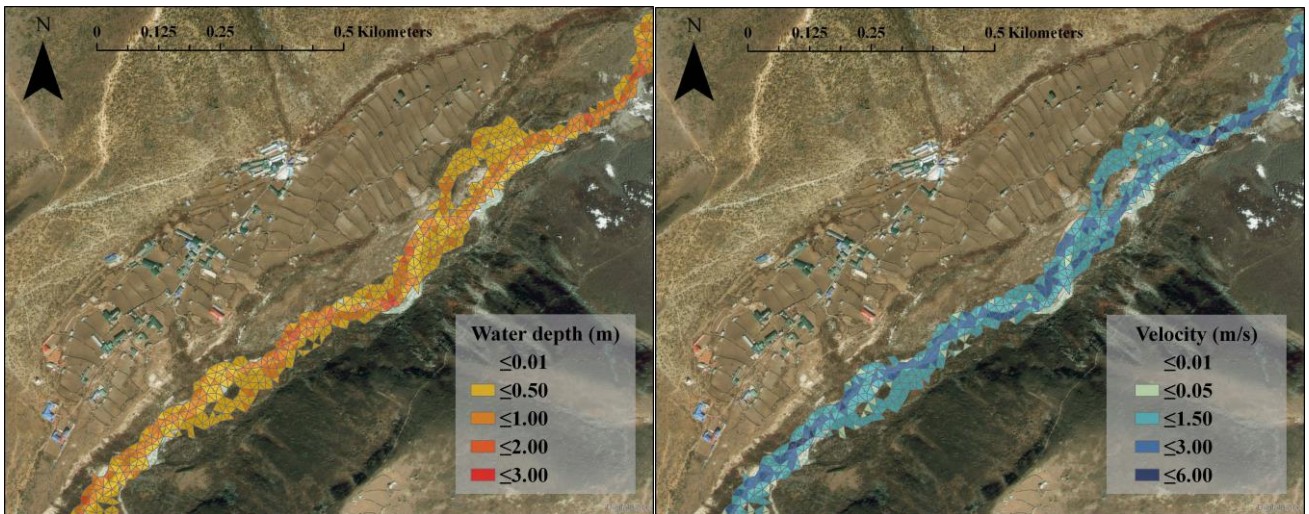

**Figure 10: Maximum water depth (left) and velocity (right) at Dingboche for the 2045 large avalanche, MPM model.**

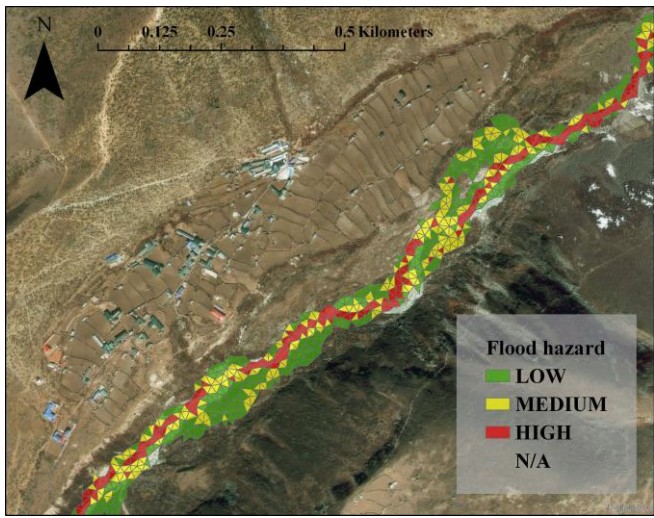

**Figure 11: Hazard level at Dingboche for the 2045 large avalanche, MPM model.**

## 4. Discussion

### 4.1 Comparison to Other Studies

To the authors' knowledge, this study is the first to model an avalanche-induced GLOF process chain at Imja Tsho. Results indicate that Imja Tsho presents little hazard from an avalanche-induced GLOF to downstream communities for the next three decades, if current trends of lake expansion continue. This seems to validate the conclusions of some earlier non-dynamic model studies which regarded the terminal moraine as a buffer (Fujita et al., 2009; Watanabe et al., 2009) and

suggested the presence of Dingboche and other villages downstream of Imja Tsho likely contributed to undue alarmism in assessing downstream hazard (Watanabe et al., 2009).

While the results from this study indicate no threat to the village of Dingboche, channel flooding still poses a small threat to humans and livestock working or grazing near the river, as well as river crossings further downstream. However, even in the worst-case scenario, flooding reached the village of Dingboche about an hour after the avalanche entered the lake, providing an ample window for warning and evacuation if water levels in the lake and river are monitored.

One reason the results of this study conflict with that of previous GLOF models of Imja Tsho (Somos-Valenzuela et al., 2015; Shrestha and Nakagawa, 2016) is this study modelled the breach of the terminal moraine based on an avalanche entering the lake as opposed to making assumptions regarding the breach or overtopping. Somos-Valenzuela et al. (2015) modelled the breach of the terminal moraine using a combination of empirical and numerical methods, but assumed the breach would be triggered by piping. While piping is theoretically possible, Imja's wide and gently-sloped moraine make this unlikely, especially when one considers Imja's moraine stability compared to other glacial lakes in the region (Fujita et al., 2013). Bajracharya et al. (2007) relied on a similar assumption about internal failure of the moraine and did not consider dynamic causes. The width of the terminal moraine also makes the failure—via overtopping of the moraine—modelled by Shrestha and Nakagawa (2016) unlikely, since even the largest avalanches considered in this study do not fully overtop the terminal moraine.

In contrast to studies that assume dam breaching from internal failures or wave overtopping, studies that relied more on geographic and geomorphic data concluded that Imja Tsho poses little imminent risk and that the lake is currently safe (Fujita et al., 2009; Watanabe et al., 2009; Rounce et al.; 2016), but that expansion up-glacier (eastward) must be monitored to continually assess the risk of mass movement into the lake. The results presented in this study indicate that even if eastward expansion continues, the lake will pose little risk for the next three decades, although regular monitoring of the terminal moraine and up-glacier mass movement trajectories will be needed to continually reassess downstream hazard.

**4.2 Modeling Techniques**

The use of BASEMENT in this study was a large improvement over previous models. BASEMENT was able to compute debris loads without requiring specification of average or maximum sediment concentrations in the flow, which is necessary for FLO-2D (Somos-Valenzuela et al. 2015) and the method of Shrestha and Nakagawa (2016), respectively. These requirements present a problem, since there are few well-documented extreme flow events from which these parameter values can be estimated (Worni et al., 2014). In contrast, geomorphic parameters needed for BASEMENT can be estimated from field data that is not event-specific. BASEMENT thus reduces the amount of data needed to run a simulation, which is a benefit in data-scarce mountain regions; the extent of necessary sensitivity analysis is also correspondingly reduced. Furthermore, BASEMENT is open-access, making it ideal for stakeholders in developing countries with limited budgets for purchasing commercial software. It also has a user-friendly GUI and can be executed on most modern desktop computers,

which facilitates knowledge transfer such that national agencies, with some help from specialists, can adapt the models to new scenarios.

Open-source software such as r.avaflow (Mergili et al., 2017) may contribute to future hazard analysis given the software's ability to model two-phase (i.e., solid debris and water) flow, once calibrated with real-world data. However, impulse wave dynamics are substantially affected by the chosen solid phase parameters in the underlying model (Pudasaini, 2014), particularly the solids concentration within the lake, which requires more data and sensitivity analysis than a water-based model. Overall, two-phase models will likely be complementary to, rather than a replacement of, process chain models, since both have advantages for different applications (Worni et al., 2014).

## 4.3 Limitations and Uncertainties

It is important to highlight that this study assessed only avalanche-induced waves as GLOF triggers and their potential for erosion of the terminal moraine and downstream inundation. Mass movement from rockfalls was not assessed, as the lateral moraines of the lake are well-developed and pose little risk of a large slope failure (Rounce et al., 2016). Previous work has considered the possibility of self-destructive moraine failure through piping, seepage, and subsequent erosion (Shrestha et al., 2013; Shrestha and Nakagawa, 2016; Somos-Valenzuela et al., 2015), which historically is the second most common cause of GLOFs in the Himalayas (Emmer and Cochachin, 2013; Falátková, 2016). The melting of buried ice within the moraine could weaken the moraine's ability to withstand the hydrostatic pressure and trigger a self-destructive failure; however, Imja's wide, gently-sloped moraine and well-developed outlet complex suggests this is unlikely. Furthermore, the results of this study indicate surface erosion from an overtopping wave will not likely reach the ice core and accelerate melting. Still, seepage at the terminal moraine has been observed on many occasions (Somos-Valenzuela et al., 2015), including the authors' most recent visit (April 2017), and should not be disregarded from future hazard assessments of the lake.

The identification of initial release areas for avalanches is perhaps the largest source of uncertainty in the work reported here. The high altitude of the Himalayas allows for avalanche ice to be frozen to the bedrock, which allows larger volumes to accumulate before release and can lead to avalanches in the millions of cubic meters (Alean, 1985). The thickness of these masses can reach up to 60 m, but a more realistic value would range from 20-45 m, based on the method of Wang et al. (2012). The large avalanches used in this study had surface areas of approximately $1.34 \times 10^5$ m$^2$, well within the range of large historical avalanches in the Swiss Alps, although these generally had smaller volumes (Alean, 1985). Therefore, the large avalanches used for this study were deemed reasonable and representative of potential extreme events that would represent a worst-case scenario.

## 4.4 Future Work

One of the major goals of this study was to create a replicable GLOF model that could be applied to other lakes besides Imja Tsho. Results suggest that lakes with larger terminal moraines (such as Thulagi and Lower Barun in Nepal) may be safer

than previously assessed, but considerable hazard may still apply to those with smaller or steeper moraines. Future work should apply this model to other lakes with smaller moraines, such as Lumding Tsho, Chamlang North Tsho, Chamlang South Tsho, and Tsho Rolpa (Rounce et al., 2016). Finally, monitoring of Imja Tsho's terminal moraine and expansion should continue, so that assumptions concerning the lake's hazard can be regularly re-evaluated.

## 5. Conclusions

The objective of this study was to model a GLOF process chain from its origin as a high mountain rock and ice slope failure to its downstream impacts, and apply the model to a case study at Imja Tsho. The steps in achieving this were threefold, namely by modeling avalanche generation and propagation, impulse wave generation and propagation, and moraine erosion and downstream flooding. Results indicated that only the largest avalanches ($6.6 \times 10^6$ m$^3$ or greater) will result in significant amounts of mass entering the lake, and even these scenarios will not pose risk for at least three decades. However, further field data on avalanches would be beneficial for calibrating model results, which was based on limited historical data.

The transfer of momentum from the avalanche to the lake was achieved by scaling the inflow hydrograph's time and discharge, allowing momentum to be changed without artificially increasing the avalanche or lake volume. A reasonable match between the BASEMENT and the Heller-Hager method results was possible, validating BASEMENT's utility as part of a GLOF model. Two morphologic scenarios were chosen: a generalized worst-case scenario for the entire region (single-grain morphology, MPM model), and a case-specific scenario unique to Imja Tsho (multiple-grain morphology, MPM-Multi model). The former yielded greater erosion of the terminal moraine of Imja Tsho, but still yielded no flooding outside the river channel at Dingboche, indicating that, most likely, the village is safe from an avalanche triggered GLOF for the next three decades. There is still a small hazard, however, for humans working and livestock grazing near the river, which indicates a need to monitor lake and river levels in real-time.

The model developed in this study can be replicated at other lakes in the greater region, many of which lack the safeguards present at Imja Tsho, such as a wide moraine complex and distance from hanging ice. Future work should address all of these concerns, so that limited aid resources can be allocated to the most cost-effective projects.

## Acknowledgments

Funding for this project was provided by the National Science Foundation under the Dynamics of Coupled Natural and Human Systems program (award no. 1516912). The authors appreciate the efforts of Jonathan Burton and Greta Wells in carrying out the bathymetric survey at Imja Tsho. In addition, Dr. Dhananjay Regmi and Himalayan Research Expeditions were instrumental in coordinating and supporting all in-country travel and provided guides, porters and field research assistance. The authors would also like to thank Owen King for contributing the DEM used in this study.

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
