# Peer review of "Modeling the Glacial Lake Outburst Flood Process Chain in the Nepal Himalaya: Reassessing Imja Tsho's Hazard"

_Hydrology and Earth System Sciences, 2017_

## Referee Comment (RC1) · S. Cook (Referee) · 6 Feb 2018

Review of "Modeling the Glacial Lake Outburst Flood Process Chain in the Nepal Himalaya: Reassessing Imja Tsho's Hazard" (hess-2017-683) by Lala et al.

Reviewer: Simon Cook, University of Dundee, UK (s.y.cook@dundee.ac.uk)

Summary

Lala et al. present results from a modelling study that assesses the possible downstream impacts of a glacial lake outburst flood (GLOF) initiated by an ice avalanche into Imja Tsho, in the Nepal Himalaya. Imja Tsho has attracted a great deal of attention

because it is a large and growing lake surrounded by steep slopes, is bounded by an ice-cored moraine that could lose integrity over time, and because it is situated just a few km upstream from the village of Dingboche. Hence, if it were to burst, some have suggested that the impacts to Dingboche could be severe. Lala et al. critically examine this possibility and come to the conclusion that the risk to downstream communities is actually much lower than found in previous studies. Overall, I think this is a generally coherent and well-executed piece of work that will be of broad interest. I do, however, have a number of suggestions and questions that I think need to be addressed first before this manuscript could be published in full.

General comments

First, the authors have made a good case here for undertaking this study, and have outlined clearly how their modelling approach compares with previous efforts. This indicates novelty in their work. However, Imja Tsho made headlines in October 2016 when the Nepalese Army were deployed to undertake some engineering work to lower the lake, and thereby reduce GLOF hazard. Whilst the authors allude to this recent change briefly in the Introduction (P2 L23), it seems that this change (reported by the BBC to be a lowering of 3m) has not been considered in this study. For example, the bathymetric survey undertaken by the authors was completed in June 2016, several months before the lake lowering. Ultimately, my question here is: to what extent does the manual lake lowering affect the relevance of your results? For example, maybe the lake lowering by the army is insignificant in the grand scheme of things because the glacier will continue to recede (as you say in section 2.2), and occupy new overdeepenings, etc. But I think there should be more mention and consideration of this recent change in the size/volume of the lake.

Second, on p16 you mention something about the depth of erosion required to meet the ice core in the moraine, which got me thinking about ice-cored moraine degradation. From your results, it seems that the lake only really becomes a worry from 2045 because of the potential interaction with ice avalanches. Your model assumes that the

ice-cored moraine remains in a steady state (i.e. same elevation as today) over the period to 2045 and beyond. Or have I misunderstood? Is there any evidence that the moraine height and composition will change as the ice core degrades with climate amelioration? Is it right to assume that the moraine complex will stay the same to 2045? What implications does that have for your modelling? A quick google revealed this study, which may be instructive: Watanabe et al (1995) Mountain Research & Development, 15, 4, 293-300. I would like to see some discussion/consideration of how moraine lowering and progressive loss of internal ice might affect its susceptibility to being breached.

Third, you have focused on ice avalanche impact into the lake, and you make mention of some situations that you haven't modelled for (e.g. piping through the moraine), which is fine. But could the same steep slopes around the lake lead to rock avalanches or landslides into the lake, even in its current (smaller) configuration? Is the area prone to seismic shaking, which could weaken the dam or increase the potential for landslide impacts?

Fourth, in the abstract and elsewhere (e.g. Discussion), you mention how this study is reproducible with open access software, etc. But the RAMMS software costs money. Is it really fair to say that any stakeholder could access these bits of software and run the models?

Finally, the writing is generally good, but there are a lot of places where there is perhaps a lack of precision. I have listed these below in the minor comments section.

Minor comments

Abstract – I think you need to mention here that the lake was manually lowered. Seems to me to be a key part of the story that is missing from your summary.

P2 L2 – capitalise 'Earth'

P2 L3 – do H-K rivers really supply $\frac{1}{4}$ of the world's people with water? I note that the

reference you use is 10 years old – a time in which the global population has grown by ∼0.5 billion people.

P2 L9 – you should perhaps cite Quincey et al (2007) in Global & Planetary Change who examined influence of surface slope and velocity on lake formation. Indeed, you should mention velocity as a factor here.

P2 L10 – you need to say that it is through coalescence that these ponds grow to become lakes. You should perhaps also cite Benn et al (2001) in J Glac.

P2 L24 – can you say something about the amount of lake lowering by the army? Depth, volume, area changes? Needs quantitative information.

Figure 1 scale bar – seems odd to have labelled divisions of 1.25 and 2.5km, and the tick marks at 0.625km spacing. Simplify.

Figure 1 caption – you should perhaps state the source of the background image and ensure you have permissions to reproduce it. Likewise, it appears that the inset is from another paper, which is cited. But do you have permission to reproduce it here?

Figure 1 – how have you determined the dark blue areas of hanging ice? Steepness threshold from a DEM? Needs to be stated.

P4 L7 – to me, 'physically modelling' something would be a laboratory experiment, which is not what you are intending to do. I think you mean something along the lines of modelling a realistic process chain, but I'll leave the wording up to you. I don't think you mean physical modelling though.

P4 L13 – suggest you replace the word 'significant', which has specific statistical connotations.

P4 L25 – this aim confuses me. "This study seeks to assess a comprehensive set of models to evaluate...". This makes it sound as though you are testing the models themselves, AND evaluating the hazard. Is that possible? Is that what you're really

doing? Or is it that you are 'employing a comprehensive set of models to evaluate GLOF risk' (or words to that effect)? The last part of the sentence is also confusing – using easily replicable methods – surely the models ARE the methods? Would it be better here to split this into two sentences, the second saying, "This model chain represents an easily replicable method...".

P5 L3. Missing full stop.

P5 L4 – I think this statement is too general. Surely it's more relevant here to replace 'climate change' with 'glacier recession and thinning', or even 'risks associated with glacier recession' or similar. Fine to mention climate change, but the impacts are very broad indeed.

P5 L14 "an environmental flow software" – awkward wording

P5 L22-5 – is this how you have identified the blue shaded areas on Fig 1? If so, I think you need to refer to Fig 1 here somewhere.

P5 L30 –perhaps you need to present this as a full range of possible volumes.

P6 L13 – was the bathymetric survey done before or after the lake lowering work by the Nepalese Army?

P6 L24 – ok, but lake has been lowered by $\sim$3m by the army.

P8 L32 – data are plural

P10 L18 – "and an area was" doesn't make sense

P11 L7 – overdeepenings of the glacier BED, not the glacier itself.

Fig 4 inset – what do these graphs show exactly? This needs to be explained in the figure caption.

P12 L9 – what do you mean by "through 2045"? In the lake modelled for 2045 and forward in time from that point? Unclear.

P12 L10-11 – Here again some imprecision. Be careful. It reads to me as though the avalanche impacts the lake in 2025, but the resultant wave-induced erosion happens in 2045, which is clearly not what you mean to say! Reword.

Fig 8 caption – I think you need to say where these cross-sections are (i.e. in a channel) and that the locations of the transects is shown in Fig 6.

P19 L24 – in this paragraph you state that you have not modelled all scenarios of GLOF initiation, which is fair enough. But should rock avalanches or landslides be included in this list? What is the likelihood of a big slope failure into the lake, even in its current configuration? Is there significant seismic hazard here, which could enhance the possibility of slope failures?

P20 L18 – to a case study at Imja Tsho

---

## Referee Comment (RC2) · W. Schwanghart (Referee) · 2 May 2018

First of all, I apologize for the delay in submitting this review.

Lala et al. present results of a study that investigates cascading processes of an avalanche-triggered GLOF. The authors report on new bathymetric data as well as a numerical simulation chain that combines the different models RAMMS and BASE-MENT. Their results represent a valuable contribution to the discourse about the risks related to an outburst of Imja Tsho and provide new insights into GLOF modelling that go beyond this case study. As such, the manuscript is worth publishing and HESS is a suitable journal. Before publication, however, there are few issues that require further

work, some of which have been addressed by the first reviewer Simon Cook.

1. The manuscript is well written. I particularly like the layout of the controversy around the hazardousness of the lake to which the study contributes. However, I think that the discussion section could pick up more of this controversy. Instead, the discussion is very much software related (advantages of BASEMENT, two-phase models (r.avaflow)) which distracts from this controversy. I suggest to restructure the discussion, possibly with subheadings, to keep focus on the controversy.

2. Some of the initial conditions related to wave propagation are unclear. Is water that spills over the moraine routed across dry terrain, or is there some initial discharge in Imja Khola? How does flood hazard change if the river is already bankfull during Monsoon season? Moreover, does the DEM cover the area down to Dingboche? Was the DEM preprocessed and hydrologically corrected? In a recent study, we have shown that hydrodynamic models are quite sensitive to pits in the DEM as they become subsequently filled during flood-wave propagation (Bricker et al., Mountain Research and Development, 37, 5-15). Is it possible that the strong attenuation of the flood wave is due this issue? Moreover, what is the hydrograph volume that leaves the lake and what is its proportion to overall lake volume. Is there some incision into the moraine dam that lowers the lake or is the hydrograph volume merely the water that overtops the dam crest?

3. I think that the differences in the Heller-Hager model and the wave heights from BASEMENT should be discussed in the discussion section. The calibration of the model using the analytical Heller-Hager model seems admissible, although it is far from elegant. Can this be overcome somehow?

Specific comments:

2, 13: Remove "catastrophic". It is the chain of events and impacts that make these events catastrophic. But per se, they are not catastrophic.

5, 17: To my knowledge, Fischer et al.'s study adresses the European alps and not the Everest Region.

6, 17: Is this truely a <4 m resolution DEM, or is it a DEM with an accuracy of ∼4 m, as stated in the referenced paper (King et al., 2017)?

8, 19: BASEMENT

13, 21: Heller-Hager

15, 4: Debris discharge: Please clarify what you mean by this term. Sediment discharge? Or sediment and water discharge combined?

Fig 8 requires labelling (A-C) of the panels.

---

## Author Comment (AC1) · 25 May 2018

*Lala et al. present results from a modelling study that assesses the possible downstream impacts of a glacial lake outburst flood (GLOF) initiated by an ice avalanche into Imja Tsho, in the Nepal Himalaya. Imja Tsho has attracted a great deal of attention because it is a large and growing lake surrounded by steep slopes, is bounded by an ice-cored moraine that could lose integrity over time, and because it is situated just a few km upstream from the village of Dingboche. Hence, if it were to burst, some have suggested that the impacts to Dingboche could be severe. Lala et al. critically examine this possibility and come to the conclusion that the risk to downstream communities is*

*actually much lower than found in previous studies. Overall, I think this is a generally coherent and well-executed piece of work that will be of broad interest. I do, however, have a number of suggestions and questions that I think need to be addressed first before this manuscript could be published in full.*

The authors thank the reviewer for his insightful and supportive comments. Responses to each are given below.

**General comments**

*First, the authors have made a good case here for undertaking this study, and have outlined clearly how their modelling approach compares with previous efforts. This indicates novelty in their work. However, Imja Tsho made headlines in October 2016 when the Nepalese Army were deployed to undertake some engineering work to lower the lake, and thereby reduce GLOF hazard. Whilst the authors allude to this recent change briefly in the Introduction (P2 L23), it seems that this change (reported by the BBC to be a lowering of 3m) has not been considered in this study. For example, the bathymetric survey undertaken by the authors was completed in June 2016, several months before the lake lowering. Ultimately, my question here is: to what extent does the manual lake lowering affect the relevance of your results? For example, maybe the lake lowering by the army is insignificant in the grand scheme of things because the glacier will continue to recede (as you say in section 2.2), and occupy new overdeepenings, etc. But I think there should be more mention and consideration of this recent change in the size/volume of the lake.*

The reviewer rightly points out that the lake lowering project is only mentioned briefly, and it is true that a significant lowering of the lake level would affect the results. Imagery from before and after the lake lowering (Figure 1 in this response, Figure 3 in revised manuscript) shows a clear ring of discoloration and decrease in area at the outlet ponds, but not in the main part of the lake, indicating that the latter may not have been lowered to the same extent. The authors decided to remain conservative in

the hazard assessment by assuming that the lake was not lowered. A paragraph has been added to section 2.2 and reads as follows: "Although the lake was subjected to a lowering project in the summer of 2016 that reportedly lowered the lake by 3 m (BBC World Service, 2016), this lowering was not accounted for in the GLOF process chain modeling as it is unclear how much the main lake was lowered based on repeat satellite imagery (Figure 3). Specifically, WorldView-2 (0.5 m; DigitalGlobe, Inc.) images of the lake's outlet complex before and after the lowering project show a clear ring of discoloration and decrease in area around the outlet ponds, but the lack of discoloration near the shore of the main lake suggests that the main lake may not have been lowered to the same extent. Hence, the GLOF process chain was modelled conservatively by not accounting for any lake lowering."

In addition, there is no conclusion as to the original surface elevation of the lake. Most studies have put it at 5010 m (Watanabe et al., 1995; Fujita et al., 2009; Somos-Valenzuela et al., 2014; Somos-Valenzuela et al., 2016), whereas others have estimated lower elevations of 5004-5008 m (Watanabe et al., 2009; Lamsal et al., 2011; Rounce et al., 2016). This study assumed an elevation of 5007 m—within the range of estimated values—and the bathymetry was burned into the regional DEM, so that the effect of the elevation is minimal. The bathymetric survey was carried out before the lake lowering project, so the model is essentially a pre-drainage model. The choice of 5007 m versus 5010 m thus has little effect on the results; to model the lake lowering, we could remove the top 3 m of water from the lake, which would set the elevation to 5004 m based on our initial assumption. This, of course, would lead to even less flooding downstream, and hence our conclusion remains the same: Imja Tsho is likely safe for the foreseeable future.

Lamsal, D., Sawagaki, T., and Watanabe, T.: Digital terrain modelling using Corona and ALOS PRISM data to investigate the distal part of Imja Glacier, Khumbu Himal, Nepal. J. of Mountain Science, 8, 390-402, 2011

*Second, on p16 you mention something about the depth of erosion required to meet*

*the ice core in the moraine, which got me thinking about ice-cored moraine degradation. From your results, it seems that the lake only really becomes a worry from 2045 because of the potential interaction with ice avalanches. Your model assumes that the ice-cored moraine remains in a steady state (i.e. same elevation as today) over the period to 2045 and beyond. Or have I misunderstood? Is there any evidence that the moraine height and composition will change as the ice core degrades with climate amelioration? Is it right to assume that the moraine complex will stay the same to 2045? What implications does that have for your modelling? A quick google revealed this study, which may be instructive: Watanabe et al (1995) Mountain Research Development, 15, 4, 293-300. I would like to see some discussion/consideration of how moraine lowering and progressive loss of internal ice might affect its susceptibility to being breached.*

It is true that the terminal moraine has changed over time; however, the rate of these changes has slowed considerably, and the width of the moraine further acts as a stabilizer. The following has been added to section 2.2: "The terminal moraine was also assumed to remain stable in the future. While there is evidence that the moraine has lowered over time (Watanabe et al., 1995), the western shoreline of the lake adjacent to the terminal moraine has remained stable since the late 1980s (Fujita et al., 2009). Furthermore, the moraine's width and gentle slope add to its stability such that degradation of the ice core or piping will not likely pose a major risk, and a wave is more likely to cut through the outlet rather than completely overtop the moraine (Rounce et al., 2016)."

*Third, you have focused on ice avalanche impact into the lake, and you make mention of some situations that you haven't modelled for (e.g. piping through the moraine), which is fine. But could the same steep slopes around the lake lead to rock avalanches or landslides into the lake, even in its current (smaller) configuration? Is the area prone to seismic shaking, which could weaken the dam or increase the potential for landslide impacts?*

Avalanches are the primary trigger of GLOFs in the Nepal Himalaya (Falátková, 2016), and we modeled very large avalanches. If these did not produce a major flood, it is unlikely that a smaller rock avalanche or landslide would trigger a GLOF, especially since the lateral moraines around the lake are well-developed and unlikely to produce large rockslides (Rounce et al., 2016). As for self-destructive failure, the width and low slope of the dam makes this even less likely than other lakes in the region. The following sentence was added to section 4.3: "Mass movement from rockfalls was not assessed, as the lateral moraines of the lake are well-developed and pose little risk of a large slope failure (Rounce et al., 2016)." Regarding seismic shaking, the 2015 earthquake and aftershock did not seem to significantly damage the terminal moraine, thus seismic shaking was disregarded. The following sentence has been added to section 2.2: "Weakening of the terminal moraine due to seismic activity was similarly disregarded based on the moraine's width and the lack of appreciable harm it suffered from the 2015 Gorkha earthquake and the earthquake's aftershocks (Byers et al., 2017)."

*Fourth, in the abstract and elsewhere (e.g. Discussion), you mention how this study is reproducible with open access software, etc. But the RAMMS software costs money. Is it really fair to say that any stakeholder could access these bits of software and run the models?*

It is true that RAMMS is not free. The abstract was changed to read "…these models were designed for ease and flexibility such that local or national agency staff with reasonable training can apply them to model the GLOF process chain for other lakes in the region." The discussion was changed to read "…BASEMENT is open-access, making it ideal for stakeholders in developing countries with limited budgets for purchasing commercial software."

*Finally, the writing is generally good, but there are a lot of places where there is perhaps a lack of precision. I have listed these below in the minor comments section.*

*Minor comments*

*Abstract – I think you need to mention here that the lake was manually lowered. Seems to me to be a key part of the story that is missing from your summary.*

"...threatening both property and human life, which prompted the Nepali government to construct outlet works to lower the lake level" added to abstract.

*P2 L2 – capitalise "Earth"*

"Earth" capitalised

*P2 L3 – do H-K rivers really supply $\frac{1}{4}$ of the world's people with water? I note that the reference you use is 10 years old – a time in which the global population has grown by ~0.5 billion people*

Citation updated to Matthew (2013), and corrected to "over a fifth of the earth's population"

*P2 L9 – you should perhaps cite Quincey et al (2007) in Global Planetary Change who examined influence of surface slope and velocity on lake formation. Indeed, you should mention velocity as a factor here.*

Sentence changed to "For glaciers where the surface slope is small and surface velocity is slow ($< 10$ m a$^{-1}$), meltwater and precipitation tend to pool in small ponds, which act as a heat sink for solar radiation and accelerate glacial melt (Quincey et al., 2007; Mertes et al., 2016)."

*P2 L10 – you need to say that it is through coalescence that these ponds grow to become lakes. You should perhaps also cite Benn et al (2001) in J Glac.*

Sentence changed to "Eventually, these small ponds can coalesce and become the large glacial lakes found throughout the Hindu Kush - Himalaya Region (Benn et al., 2012)."

[Figure]

*P2 L24 – can you say something about the amount of lake lowering by the army? Depth, volume, area changes? Needs quantitative information.*

It is currently unclear how much the main part of the lake was actually lowered (see response to Comment 1 above)

*Figure 1 scale bar – seems odd to have labelled divisions of 1.25 and 2.5km, and the tick marks at 0.625km spacing. Simplify*

Scale bar changed to 1 km divisions

*Figure 1 caption – you should perhaps state the source of the background image and ensure you have permissions to reproduce it. Likewise, it appears that the inset is from another paper, which is cited. But do you have permission to reproduce it here?*

Citation for DigitalGlobe, Inc. added to caption for background imagery. Regarding the inset, the authors requested permission from the publisher, but the publisher is not the copyright holder of the figure. A similar, open-access figure (Nicholson et al., 2016; Creative Commons Attribution International License) was used as a replacement.

Nicholson, K., Hayes, E., Neumann, K., Dowling, C., and Sharma, S.: Drinking water quality in the Sagarmatha National Park, Nepal. J. of Geosciences and Environment Protection, 4, 43-54, 2016

*Figure 1 – how have you determined the dark blue areas of hanging ice? Steepness threshold from a DEM? Needs to be stated.* Citation to Rounce et al. (2016) added to caption to clarify source of hanging ice data. Yes, hanging ice is determined from Landsat imagery and a steepness threshold from a DEM (Section 2.1 explains this in more detail).

*P4 L7 – to me, 'physically modelling' something would be a laboratory experiment, which is not what you are intending to do. I think you mean something along the lines of modelling a realistic process chain, but I'll leave the wording up to you. I don't think you mean physical modelling though.* "physically modeling it" changed to "modeling it

through a realistic process chain"

*P4 L13 – suggest you replace the word 'significant', which has specific statistical connotations.* "significant" changed to "high"

*P4 L25 – this aim confuses me. "This study seeks to assess a comprehensive set of models to evaluate. . .". This makes it sound as though you are testing the models themselves, AND evaluating the hazard. Is that possible? Is that what you're really doing? Or is it that you are 'employing a comprehensive set of models to evaluate GLOF risk' (or words to that effect)? The last part of the sentence is also confusing – using easily replicable methods – surely the models ARE the methods? Would it be better here to split this into two sentences, the second saying, "This model chain represents an easily replicable method...".*

Sentence changed to "This study seeks to employ a comprehensive set of models to evaluate the present and future hazard associated with avalanche-generated impulse waves at Imja Tsho. This model chain represents an easily replicable method that can be applied to other lakes."

*P5 L3. Missing full stop.*

Full stop added

*P5 L4 – I think this statement is too general. Surely it's more relevant here to replace 'climate change' with 'glacier recession and thinning', or even 'risks associated with glacier recession' or similar. Fine to mention climate change, but the impacts are very broad indeed.*

Sentence changed to "Understanding these components will assist in the wider goal of helping local communities adapt to the risks associated with glacier recession, increasing the capacity for climate change resilience."

*P5 L14 "an environmental flow software" – awkward wording*

[Figure]

Changed to "environmental flow modeling software"

*P5 L22-5 – is this how you have identified the blue shaded areas on Fig 1? If so, I think you need to refer to Fig 1 here somewhere.*

Reference to Fig. 1 added.

*P5 L30 –perhaps you need to present this as a full range of possible volumes.*

Sentence changed to "...the total avalanche volume could reach from $2.7 \times 10^4$ to $6.7 \times 10^6$ m3 (Rounce et al., 2016)."

*P6 L13 – was the bathymetric survey done before or after the lake lowering work by the Nepalese Army.*

The bathymetric survey was done before the lowering project (see response to Comment 1 above). The month of the lowering project (November) was added to p. 2 to clarify this.

*P6 L24 – ok, but lake has been lowered by ~3m by the army.*

See response to Comment 1. There is no evidence that the main part of the lake was actually lowered to the same extent as the outlet ponds, and assuming no lowering provides a more conservative estimate of hazard.

*P8 L32 – data are plural*

"is" changed to "are"

*P10 L18 – "and an area was" doesn't make sense.*

"was" changed to "of"

*P11 L7 – overdeepenings of the glacier BED, not the glacier itself.*

"bed" added after "glacier"

*Fig 4 inset – what do these graphs show exactly? This needs to be explained in the*

*figure caption.*

The graphs show a time series of avalanche material entering the lake's estimated 2045 boundaries. Caption changed to "Figure 5: Deposition from the two large avalanche scenarios superimposed over estimated future lake extents, and time series of avalanche material entry into the lake for the 2045 estimated lake extents (inset). The northeast avalanche enters perpendicularly to the lake expansion trajectory, whereas the southeast avalanche enters with an almost direct trajectory toward the terminal moraine."

*P12 L9 – what do you mean by "through 2045"? In the lake modelled for 2045 and forward in time from that point? Unclear.*

"through" changed to "at or before"

*P12 L10-11 – Here again some imprecision. Be careful. It reads to me as though the avalanche impacts the lake in 2025, but the resultant wave-induced erosion happens in 2045, which is clearly not what you mean to say! Reword.*

Sentence changed to "Specifically, the large avalanche from the northeast reached the lake at the 2025 predicted extent, but mass entry was too small and failed to produce measurable erosion at the terminal moraine unless the predicted 2045 lake was simulated."

*Fig 8 caption – I think you need to say where these cross-sections are (i.e. in a channel) and that the locations of the transects is shown in Fig 6.*

"...at the lake outlet (A), moraine outlet (B), and Imja Khola channel (C) transects show in Figure 7" added to caption. Please note that Figure 8 is now Figure 9 due to the addition of a new figure (Figure 3).

*P19 L24 – in this paragraph you state that you have not modelled all scenarios of GLOF initiation, which is fair enough. But should rock avalanches or landslides be included in this list? What is the likelihood of a big slope failure into the lake, even in its*

*current configuration? Is there significant seismic hazard here, which could enhance the possibility of slope failures?*

See response to Comment 3. "Mass movement from rockfalls was not assessed, as the lateral moraines of the lake are well-developed and pose little risk of a large slope failure (Rounce et al., 2016)." added to paragraph.

*P20 L18 – to a case study at Imja Tsho.*

sentence changed to "...to a case study at Imja Tsho"

14 May 2016

100m

29 Oct 2016

**Fig. 1.** WorldView-2 (0.5 m; DigitalGlobe, Inc.) imagery of the lake before (14 May 2016) and after (29 October 2016) the lake lowering project

---

## Author Comment (AC2) · 25 May 2018

*First of all, I apologize for the delay in submitting this review.*

*Lala et al. present results of a study that investigates cascading processes of an avalanche-triggered GLOF. The authors report on new bathymetric data as well as a numerical simulation chain that combines the different models RAMMS and BASEMENT. Their results represent a valuable contribution to the discourse about the risks related to an outburst of Imja Tsho and provide new insights into GLOF modelling that go beyond this case study. As such, the manuscript is worth publishing and HESS is a suitable journal. Before publication, however, there are few issues that require further*

[Figure]

*work, some of which have been addressed by the first reviewer Simon Cook.*

The authors would like to thank the reviewer for his supportive comments. Responses to each are given below.

*1. The manuscript is well written. I particularly like the layout of the controversy around the hazardousness of the lake to which the study contributes. However, I think that the discussion section could pick up more of this controversy. Instead, the discussion is very much software related (advantages of BASEMENT, two-phase models (r.avaflow)) which distracts from this controversy. I suggest to restructure the discussion, possibly with subheadings, to keep focus on the controversy.*

The discussion was restructured with subheadings, and the following was added under "4.1 Comparison to Other Studies:"

"One reason the results of this study conflict with that of previous GLOF models of Imja Tsho (Somos-Valenzuela et al., 2015; Shrestha and Nakagawa, 2016) is this study modelled the breach of the terminal moraine based on an avalanche entering the lake as opposed to making assumptions regarding the breach or overtopping. Somos-Valenzuela et al. (2015) modelled the breach of the terminal moraine using a combination of empirical and numerical methods, but assumed the breach would be triggered by piping. While piping is theoretically possible, Imja's wide and gently-sloped moraine make this unlikely, especially when one considers Imja's moraine stability compared to other glacial lakes in the region (Fujita et al., 2013). Bajracharya et al. (2007) relied on a similar assumption about internal failure of the moraine and did not consider dynamic causes. The width of the terminal moraine also makes the failure—via overtopping of the moraine—modelled by Shrestha and Nakagawa (2016) unlikely, since even the largest avalanches considered in this study do not fully overtop the terminal moraine.

In contrast to studies that assume dam breaching from internal failures or wave overtopping, studies that relied more on geographic and geomorphic data concluded that Imja Tsho poses little imminent risk and that the lake is currently safe (Fujita et al.,

2009; Watanabe et al., 2009; Rounce et al.; 2016), but that expansion up-glacier (east-ward) must be monitored to continually assess the risk of mass movement into the lake. The results presented in this study indicate that even if eastward expansion continues, the lake will pose little risk for the next three decades, although regular monitoring of the terminal moraine and up-glacier mass movement trajectories will be needed to continually reassess downstream hazard."

*2. Some of the initial conditions related to wave propagation are unclear. Is water that spills over the moraine routed across dry terrain, or is there some initial discharge in Imja Khola? How does flood hazard change if the river is already bankfull during Mon- soon season? Moreover, does the DEM cover the area down to Dingboche? Was the DEM preprocessed and hydrologically corrected? In a recent study, we have shown that hydrodynamic models are quite sensitive to pits in the DEM as they become subsequently filled during flood-wave propagation (Bricker et al., Mountain Research and Development, 37, 5-15). Is it possible that the strong attenuation of the flood wave is due this issue? Moreover, what is the hydrograph volume that leaves the lake and what is its proportion to overall lake volume. Is there some incision into the moraine dam that lowers the lake or is the hydrograph volume merely the water that overtops the dam crest?*

Because the initial discharge of the Imja Khola is small, it is not taken as an initial condition in the model, although there is some discharge present once the model is running due to the difference in elevation between the lake surface and the lake outlet. Even during the monsoon season, however, discharge in the Imja Khola at Dingboche is only around 4-6 $m^3$ $s^{-1}$ (Rajkarnikar, 2013)—less than 4% of the peak discharge from the GLOF flood. Dingboche sits on a terrace ~10-20 m above the river bed, and the river is never bankfull even during the monsoon season. For this reason, monsoon discharge was assumed to have a negligible effect on GLOF flood hazard. The following was added to section 2.3.4:

"Initial discharge from Imja Tsho was assumed to be negligible, since peak monsoon

Interactive
comment

discharge at Dingboche of 4-6 m$^3$ s$^{-1}$ was less than 4% of the peak discharge from the GLOF flood wave (Rajkarnikar, 2013; see Results, Figure 8)."

The DEM covers the entire area of the simulation; only the bathymetry came from a different source (field survey). The DEM was preprocessed through outlier filtering (any value exceeding ±120 m, or 3 standard deviations difference from a SRTM DEM used as a reference; see King et al., 2017), but it was not specifically hydrologically corrected. However, the DEM was converted from a raster to a triangulated irregular network (TIN) for use in the BASEMENT simulation, which acted as a smoothing method. The few small raster sinks in the floodplain (four in the inundated areas between the outlet and Dingboche, none greater than 150 m$^2$ in area) were thus effectively filled.

For a large avalanche in 2045, the volume of water leaving the lake is negligible compared to the total lake volume, suggesting that catastrophic damage to the moraine is unlikely. The moraine was not fully overtopped and hence discharge was funneled through the outlet channel of the moraine. The following sentences were added to section 3.3:

"In the first 2000 s of the simulation (i.e., before discharge lowers to ~5 m$^3$ s$^{-1}$), the total volume of water leaving the lake was approximately 251,000 m$^3$ for the MPM simulation and 166,000 m$^3$ MPM-Multi simulation—less than 0.3% of the total present lake volume (88 million m$^3$). This is notably less than the amount of avalanche material entering the lake (approximately 720,000 m$^3$; Table 5). Moreover, the lake's surface elevation remains slightly above its original elevation at the end of the simulation period (by approximately 0.25 m), suggesting that erosion of the moraine was not sufficient to allow the lake to drain quickly, and may have even allowed the lake to store more water."

"In both cases, the moraine was not fully overtopped, and erosion was confined to the outlet channel."

*3. I think that the differences in the Heller-Hager model and the wave heights from*

*BASEMENT should be discussed in the discussion section. The calibration of the model using the analytical Heller-Hager model seems admissible, although it is far from elegant. Can this be overcome somehow?*

The Heller-Hager model was used as a calibration model because it produced wave heights similar to that of FLOW3D, which was shown by Somos-Valenzuela et al. (2016) to be a more robust approach to modeling lake wave dynamics since it is not as susceptible to wave attenuation, which is present in 2-D shallow water equation (SWE) models like BASEMENT. One of the goals of this study, however, is to simply the process outlined by Somos-Valenzuela et al. (2016) by modeling both the lake wave and the downstream impacts with BASEMENT, rather than using the more complex and computationally intensive FLOW3D model for the lake wave. Since BASEMENT is a 2-D SWE model, the lake wave heights are susceptible to strong attenuation that would lessen the simulated impact downstream. Therefore, to account for this strong attenuation, the wave heights from the Heller-Hager model were used to adjust the BASEMENT simulations. The following was added in section 2.3.1 to clarify the use of the Heller-Hager method:

"...it has been used to successfully model some real-world events and performs well in characterizing the impulse wave within the lake, which makes it a useful as a calibration measure for more complex hydrodynamic models (Somos-Valenzuela et al., 2016). Moreover, it is not as susceptible to wave attenuation inherent in 2-D SWE models such as BASEMENT, making it an ideal calibration measure that is both simple and accurate."

***Specific comments***

*2, 13: Remove "catastrophic". It is the chain of events and impacts that make these events catastrophic. But per se, they are not catastrophic.*

"catastrophic" removed

*5, 17: To my knowledge, Fischer et al.'s study adresses the European alps and not the Everest Region.*

Noted. Statement has been deleted.

*6, 17: Is this truly a <4 m resolution DEM, or is it a DEM with an accuracy of ~4 m, as stated in the referenced paper (King et al., 2017)?*

The resolution of the DEM is 3.57 m, but to maintain consistency with the text of King et al. (2017) the manuscript was corrected to "~4 m"

*8, 19: BASEMENT*

corrected to "BASEMENT"

*13, 21: Heller-Hager*

corrected to "Heller-Hager"

*15, 4: Debris discharge: Please clarify what you mean by this term. Sediment discharge? Or sediment and water discharge combined?*

Clarified to "Combined sediment and water discharge;" clarification also made in Figure 8 caption.

*Fig 8 requires labelling (A-C) of the panels.*

Figure 8 panels labeled to match that of Fig. 7 (Figures 9 and 8, respectively, in revised manuscript)
* * *